

# Characterizing the hadronization of parton showers using the HOMER method

Benoît Assi[1⋆], Christian Bierlich[2†], Philip Ilten[1‡], Tony Menzo[1∘], Stephen Mrenna[1,3§],
Manuel Szewc[1,4¶], Michael K. Wilkinson[1∥], Ahmed Youssef[1♠] and Jure Zupan[1♩]

**1** Department of Physics, University of Cincinnati, Cincinnati, Ohio 45221, USA
**2** Department of Physics, Lund University, Box 118, SE-221 00 Lund, Sweden
**3** Computational Science and AI Directorate, Fermilab, Batavia, Illinois, USA
**4** International Center for Advanced Studies (ICAS), ICIFI and ECyT-UNSAM,
25 de Mayo y Francia, (1650) San Martín, Buenos Aires, Argentina

⋆ assibt@ucmail.uc.edu , † christian.bierlich@hep.lu.se , ‡ philten@cern.ch ,
∘ menzoad@mail.uc.edu , § mrenna@fnal.gov , ¶ szewcml@ucmail.uc.edu ,
∥ michael.wilkinson@uc.edu , ♠ youssead@ucmail.uc.edu , ♩ zupanje@ucmail.uc.edu

## Abstract

We update the HOMER method, a technique to solve a restricted version of the inverse problem of hadronization – extracting the Lund string fragmentation function $f(z)$ from data using only observable information. Here, we demonstrate its utility by extracting $f(z)$ from synthetic PYTHIA simulations using high-level observables constructed on an event-by-event basis, such as multiplicities and shape variables. Four cases of increasing complexity are considered, corresponding to $e^+e^-$ collisions at a center-of-mass energy of 90 GeV producing either a string stretched between a $q$ and $\bar{q}$ containing no gluons; the same string containing one gluon $g$ with fixed kinematics; the same but the gluon has varying kinematics; and the most realistic case, strings with an unrestricted number of gluons that is the end-result of a parton shower. We demonstrate the extraction of $f(z)$ in each case, with the result of only a relatively modest degradation in performance of the HOMER method with the increased complexity of the string system.



## Contents



# 1   Introduction

Recently, new machine learning (ML) techniques have been developed to model hadronization – the process in which hadrons are formed from their quark and gluon constituents. In particular, the MLHAD [1–3] and HADML [4–6] collaborations have been developing ML-based frameworks to improve the description of hadronization in Monte-Carlo simulations of particle collisions, such as the ones used in PYTHIA [7] or in HERWIG [8].

The current strategy of the MLHAD collaboration is to use the PYTHIA Lund string fragmentation model as a base model that a more flexible ML-based modification can then augment. The first goal of this strategy was to solve a restricted version of the inverse problem for hadronization, i.e., to work entirely within the Lund string fragmentation model and then extract the Lund-string fragmentation function $f(z)$ from data without requiring an explicit parametric form for $f(z)$. This was first demonstrated in ref. [3] for the simplified case of $q\bar{q}$ strings using the HOMER method. Here, we improve the HOMER method to extract $f(z)$ in the more realistic scenario of strings with an arbitrary number of gluons and demonstrate the technique using synthetic data. This is a necessary step that should be sufficient for HOMER to be used to extract $f(z)$ directly from actual experimental data.

This paper is structured as follows. In section 2 we review the treatment of gluons in the Lund string fragmentation model, while in section 3 we introduce the necessary modifications to the HOMER method. Section 4 contains numerical results, while section 5 contains conclusions and future outlook. The appendices contain further useful details; in appendix A, we illustrate the necessary HOMER modification for a toy example; appendix B contains additional results on $f(z)$ extraction from synthetic data; while appendix C contains a study where for generation of synthetic data multiple parameters were varied.

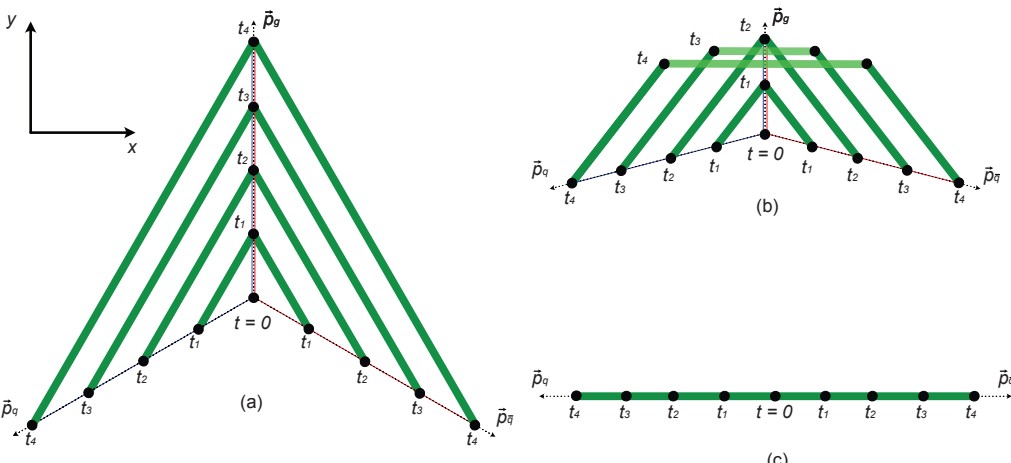

Figure 1: A string system at times $t = \{0, t_1, t_2, \ldots\}$, in three different configurations. The quark and anti-quark have momenta $\vec{p}_q$ and $\vec{p}_{\bar{q}}$, respectively. (a) A gluon with momentum $\vec{p}_g$ has enough energy that it will not be lost to the string before the system hadronizes. (b) The gluon loses all energy at a time between $t_2$ and $t_3$, resulting in a third string region in light green, parallel to the $q\bar{q}$ axis. (c) In the limit $E_g \to 0$ the gluon kink vanishes, and we are left with a normal $q\bar{q}$ string.

## 2 Lund string fragmentation model for strings with gluons

In PYTHIA, color-singlet combinations of partons are transformed into hadrons using the Lund string fragmentation model [9, 10]. In the simplest case, the string has only a $q$ and a $\bar{q}$ as endpoints with no additional gluons. The presence of gluons introduces several complications to the hadronization model that will be addressed below.

Consider first the algorithm for the hadronization of the simple $q_i\bar{q}_i$ string. The flavor selection of the $q_j\bar{q}_j$ pair that is to be created by the string break is followed by the selection of the transverse momentum of the $q_j\bar{q}_j$ pair $\Delta\vec{p}_T = (\Delta p_x, \Delta p_y)$. This is sampled from a normal distribution, whose width is given by an adjustable phenomenological parameter $\sigma_T/\sqrt{2}$. The emitted hadron receives a fraction $z$ of the lightcone momentum of the string with $z$ sampled from the symmetric Lund fragmentation function[1]

$$f(z) \propto \frac{(1-z)^a}{z} \exp\left(-\frac{bm_T^2}{z}\right), \tag{1}$$

where $m_T^2 \equiv m_{ij}^2 + p_T^2$ is the square of the transverse mass, $m_{ij}$ is the mass of the emitted hadron, and $a$ and $b$ are fixed model parameters determined from fits to data. The transverse momentum of the emitted hadron $\vec{p}_T$ is constructed as the combined $\vec{p}_T$ of the two quarks producing the hadron. If the hadron is the result of two neighboring string breaks $i$ and $j$, then $\vec{p}_T$ is the vector sum of $\vec{p}_{T,i}$ and $\vec{p}_{T,j}$. The endpoint hadrons contain the endpoint quarks, which have no $\vec{p}_T$, since the hadronization model is calculated in the string rest frame.

To better understand the effect of including gluons, it is instructive to first study the $q\bar{q}$ system in the 1+1 dimensional case, with coordinates $x$ and $t$ for space and time. There, the string follows simple equations of motion given by the differential equations:

$$\frac{dp}{dt} = \frac{dE}{dx} = -\kappa, \tag{2}$$

---

[1]Note that eq. (1) does not contain a normalization factor, which is required for $f(z)$ to be a probability distribution. In PYTHIA, the distribution is normalized by its maximum, which can be calculated analytically.

where $p$ and $E$ are momentum and energy, respectively, and $\kappa$ is the string tension, with a numerical value of roughly 1 GeV/fm. The initial conditions of this system in *e.g.*, $e^+e^- \to Z \to q\bar{q}$, has the quark and anti-quark starting at a single point, and then moving away from each other back-to-back. As they move, the string will stretch, and the quark and anti-quark will lose energy and momentum to the string. If the total energy of the system ($E_{\text{cm}}$) is so small that the string does not break, the $q\bar{q}$ pair will lose all their energy and turn around at $t = E_{\text{cm}}/(2\kappa)$. The motion will continue indefinitely, and is known as the so-called yo-yo-mode of the string.

The basic system considered above does not include gluons. They are introduced as so-called kinks on the string, and therefore depart from the 1+1 dimensional approximation. In the following we will give a brief introduction to how gluons are introduced into the model, see refs. [10–12] for more details. String systems can become very complex. In a realistic collision, one must handle string loops, junctions[2] and multi-junction topologies as well as gluons on simple strings. The most practical way of handling complex topologies is to reduce them to a system already known, *i.e.*, a chain of color/anti-color pairs.

A good heuristic is given by a high-energy $qg\bar{q}$ system, where all constituents are hard enough that they will only lose a small fraction of energy before hadronization.[3] In this case, the string system is as depicted in fig. 1(a). The color of the quark connects to the anti-color of the gluon. The color of the gluon, in turn, connects to the anti-color of the anti-quark. With the system starting at a single point at time $t = 0$, the system will evolve to the $t = t_4$ configuration and then, finally, hadronize, where we take $t_4$ to be the hadronization time of this particular string. The characteristic, measurable feature of the system, is that the regions between quark, anti-quark, and gluon are filled with hadrons.

A slightly more complicated system is offered in fig. 1(b). Here the gluon is soft, and will therefore transfer all its energy to the string, before the system hadronizes. In the situation where the gluon energy is lost, but the $q\bar{q}$ pair continues outwards, a new string region will emerge in place of the gluon. The two new so-called pseudo-kinks depicted in the figure do not correspond to two new gluons, but are emerging only due to the original gluon losing its energy. The new string piece moves according to eq. (2) and will therefore be pulled down towards a reference line, not shown in the schematic, connecting the quark and anti-quark. Since the energy of the original gluon is lost to the string, the string pieces connecting the new piece to the quark and the anti-quark, cannot grow [13]. Finally we show, in fig. 1(c) the limit $E_g \to 0$. The physical gluon is now gone, and we recover the normal $q\bar{q}$ string.

Both systems in fig. 1 (a) and (b) can be treated as a collection of two or three smaller string regions equivalent to $q\bar{q}$ strings, apart from more detailed considerations, such as the fact that a hadron can be formed over two string regions. In this case, a hadron is created by a string piece from two (or more) adjacent string regions. This can happen both across a kink made by a real gluon with energy and momentum, fig. 1(a), as well as string regions created by gluons which have lost their energy to the string, fig. 1(b). In the literature this is known as hadrons being produced around a corner.

Being able to reduce complex topologies to ones made of simple straight strings, implies that the hadronization of a $qg\bar{q}$ string in the Lund string model still depends on the same parameters as the hadronization of a $q\bar{q}$ string. That is, modeling the hadronization still consists of sampling (1) the endpoint from which the fragmentation occurs, (2) the flavor of the emitted hadron, (3) the transverse momentum kick $\Delta\vec{p}_{\text{T}}$, and (4) the light-cone momentum fraction $z$. It is only the deterministic relationship between the sampled random variables and the hadron four-momenta that now becomes more complicated. Additional PYTHIA parameters such as `FragmentationSystems:mJoin`, the minimum quark-gluon mass below which

---

[2]A junction is a node connecting three quarks or anti-quarks.

[3]As a by-product of eq. (1) one can obtain the hadronization time as a function of the parameters $a$ and $b$.

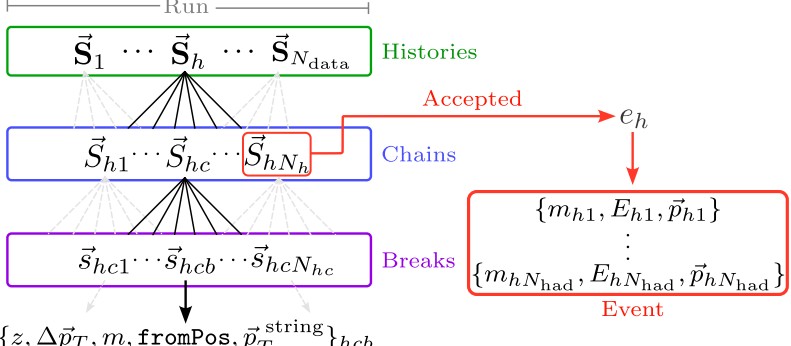

$$\{z, \Delta\vec{p}_T, m, \texttt{fromPos}, \vec{p}_T^{\text{string}}\}_{hcb}$$

Figure 2: Schematic from ref. [3], detailing different components of a simulated run. String breaks are grouped into fragmentation chains, while collections of rejected and accepted fragmentation chains form fragmentation histories. Observable events are obtained from the last, accepted fragmentation chain. A collection of multiple events is a run.

the gluon four-momentum is absorbed into the closest endpoint, are introduced for numerical convenience, but do not modify the essential probabilistic dependencies of the model.

The string hadronization of both $q\bar{q}$ and $qg\bar{q}$ string hadronization also pass through a `finalTwo` filter, which ensures that a final pair of hadrons can be produced that conserve energy and momentum and are consistent with the string area law. Hadronization chains that do not pass the filter are restarted using the same initial conditions but with a new pseudo-random number seed. This process can be repeated several times until the `finalTwo` step is successful.

Since the HOMER method does not change the two essential elements of the hadronization–the string break mechanism and the final filter – this method can be applied almost unchanged to the case of strings with additional gluons. In particular, the data structure passed to HOMER and the notation used is unchanged, see Figure 2. The fundamental object is the **string break** given by a seven-dimensional vector

$$\vec{s}_{hcb} = \left\{z, \Delta\vec{p}_T, m, \texttt{fromPos}, \vec{p}_T^{\text{string}}\right\}_{h,c,b}, \tag{3}$$

where $z$ is the light-cone momentum fraction of the hadron, $\Delta\vec{p}_T = (\Delta p_x, \Delta p_y)$ is the two-dimensional momentum kick of the emitted hadron, $m$ is the hadron mass, the parameter `fromPos` encodes whether the string break occurred at the positive or the negative end of the string, and $\vec{p}_T^{\text{string}} = (p_x^{\text{string}}, p_y^{\text{string}})$ is the transverse momentum of the string before the breaking. A sequence of string breaks forms a **fragmentation chain**

$$\vec{S}_{hc} = \left\{\vec{s}_{hc1}, \ldots, \vec{s}_{hcN_{h,c}}\right\}. \tag{4}$$

A vector of rejected fragmentation chains $\vec{S}_{h1}, \ldots, \vec{S}_{hN_h-1}$ and the accepted fragmentation chain $\vec{S}_{hN_h}$ then form an **internal simulation history**

$$\mathbf{S}_h = \left\{\vec{S}_{h1}, \ldots, \vec{S}_{hN_h}\right\}. \tag{5}$$

Here, $h = 1, \ldots, N_{\text{data}}$ is the simulation history index, with $N_{\text{data}}$ the total number of events which are collected in a **run**; $c = 1, \ldots, N_h$ is the fragmentation chain index for a particular simulation history of index $h$, which has $N_h - 1$ rejected fragmentation chains and one accepted fragmentation chain; while $b = 1, \ldots, N_{h,c}$ is the string break index that runs over a fragmentation chain of index $c$ with a total of $N_{h,c}$ string breaks.

A measurable **event** $e_h$ is fully determined by the accepted fragmentation chain, $\vec{S}_{hN_h}$,

$$e_h \equiv e\left(\vec{\mathbf{S}}_h\right) \equiv e\left(\vec{S}_{hN_h}\right), \tag{6}$$

where $e_h$ is an unordered list of $N_{\text{had}} = N_{h,N_h} + 2$ laboratory frame four momenta $(E_i, \vec{p}_i)$ and masses $m_i$ of the produced hadrons, possibly including additional information related to the flavor composition of the hadrons, such as their charge,

$$e_h = \left\{\{m_{h1}, E_{h1}, \vec{p}_{h1}\}, \ldots, \{m_{hN_{\text{had}}}, E_{hN_{\text{had}}}, \vec{p}_{hN_{\text{had}}}\}\right\}. \tag{7}$$

While the accepted fragmentation chain, $\vec{S}_{hN_h}$, fully determines the event $e_h$, the reverse is not true; several different fragmentation chains can result in the same[4] event. That is, the four momenta in eq. (7) are calculated from the accepted string fragmentation chain quantities $\vec{S}_{hN_h}$ by boosting the momenta of the produced hadrons to the laboratory frame. The reverse is not possible; the observed four momenta of hadrons do not uniquely determine the accepted fragmentation chain $\vec{S}_{hN_h}$, making it more challenging to find the solution to the inverse problem for hadronization, especially for strings with many gluons. Our approach to finding the solution requires some modifications to the original HOMER method [3], which are introduced in the next section. Empirically, we observe that the impact of degeneracy depends on the choice of observables and the statistical size of the samples considered.

## 3 The modified HOMER method

The HOMER method consists of three steps, which we review in the following subsection before introducing in section 3.2 the modifications in Step 2 that are necessary for the case of strings with gluons. Note that Step 1 and Step 3 remain unchanged from ref. [3]. For a diagramatic summary of the HOMER method used in ref. [3] we refer the reader to fig. 2 in ref. [3], to be compared with the modified Step 2 shown in fig. 3.

### 3.1 Review of the HOMER method

The goal of the HOMER method is to extract the fragmentation function $f_{\text{HOMER}}(z)$ that best describes hadronization data; in our case, these are synthetic data produced using $f_{\text{data}}(z)$. The starting point is a **baseline simulation model** that uses the fragmentation function $f_{\text{sim}}(z)$. The HOMER method finds the appropriate weights which transform $f_{\text{sim}}(z)$ into $f_{\text{HOMER}}(z) \approx f_{\text{data}}(z)$.

This is achieved by splitting the problem into three steps. In Step 1, a classifier is trained to distinguish between simulated events and data using the standard binary cross-entropy (BCE) loss function using measurable observables $\vec{x}_h$ as inputs. The output of a classifier $y(\vec{x}_h) \in [0,1]$ gives event-level weights

$$w_{\text{class}}(e_h) \equiv \frac{y(\vec{x}_h)}{1 - y(\vec{x}_h)}. \tag{8}$$

These are estimators of the exact weights, *i.e.*, the ratios of probabilities for an event $e_h$ in the exact hadronization model and the baseline simulation model,

$$w_{\text{class}}(e_h) \approx w_{\text{exact}}(e_h) \equiv p_{\text{exact}}(e_h)/p_{\text{sim}}(e_h).$$

---

[4]In this work, we consider two events to be the same if they are identical up to arbitrarily small numerical differences in the continuous features.

If needed, the exact event-level weights can be derived from the history-level exact weights $w_{\text{exact}}(\vec{\mathbf{S}}_h)$, which contain all fragmentation-level information, see refs. [3,14]. In the rest of the manuscript, we rely on these history-level exact weights to provide the optimal HOMER performance. In contrast, we never explicitly compute the event-level exact weights, relying instead on $w_{\text{class}}$ to quantify the sensitivity to hadronization of a particular choice of observables $\vec{x}_h$. The term *exact weights* thus always refers to history-level weights (which may be marginalized over unseen information) and never to the explicitly computed event-level weights.

In Step 2, the weights for each string break $w_s^{\text{infer}}(\vec{s}_{hcb})$ are extracted from event-level weights $w_{\text{class}}(e_h)$, allowing us to infer new probabilities for each string break by reweighting the baseline probabilities $p_{\text{sim}}(\vec{s}_{hcb})$ to

$$p_{\text{infer}}(\vec{s}_{hcb}) = w_s^{\text{infer}}(\vec{s}_{hcb}) p_{\text{sim}}(\vec{s}_{hcb}), \tag{9}$$

such that combining probabilities for all fragmentations in a simulation of the event gives the best approximation of $w_{\text{class}}(e_h)$.

To extract the individual string break weights $w_s^{\text{infer}}(\vec{s}_{hcb})$ from the event-level weights $w_{\text{class}}(e_h)$ we need to relate them explicitly. However, this faces two complications. First, the PYTHIA simulation history contains string fragmentation chains rejected by the `finalTwo` filter. Second, different fragmentation chains can give rise to the same observable event, and are thus physically indistinguishable. That is, the event-level weight $w_{\text{class}}(e_h)$, which is an observable quantity, corresponds to an average over fragmentation chain weights that are themselves products of individual string break weights $w_{\text{infer}}(\vec{S}_{hc}) = \prod_{b=1}^{N_{hc}} w_s^{\text{infer}}(\vec{s}_{hcb})$, which reweight the results of a simulation. Combining the two effects, the event-level weight $w_{\text{infer}}(e_h) \approx w_{\text{class}}(e_h)$ can be calculated as

$$w_{\text{infer}}(e_h) = \frac{p_{\text{sim}}^{\text{acc}}}{p_{\text{infer}}^{\text{acc}}} \big\langle w_{\text{infer}}(\vec{S}_{jN_j}) \big\rangle_{e_j = e_h}, \tag{10}$$

where the average is taken over all accepted fragmentation chains that lead to the same observable event $e_h$; $p_{\text{sim}}^{\text{acc}} = N_{\text{acc}}/N_{\text{tot}}$ is the `finalTwo` efficiency for the baseline hadronization model; $N_{\text{acc}} = N_{\text{sim}}$ is the number of accepted fragmentation chains in the simulation with $N_{\text{sim}}$ events; $N_{\text{tot}}$ is the total number of chains in the fragmentation history, including the rejected ones; and

$$p_{\text{infer}}^{\text{acc}} = \frac{\sum_{\vec{S}_{jk} \in \{\vec{s}_{hc}^{\text{acc}}\}} w_{\text{infer}}(\vec{S}_{jk})}{\sum_{\vec{S}_{jk} \in \{\vec{S}_{hc}\}} w_{\text{infer}}(\vec{S}_{jk})}, \tag{11}$$

where the sum in the numerator (denominator) is over accepted (all) simulated fragmentation chains, is the equivalent `finalTwo` efficiency for reweighted string break probabilities given by eq. (9).

When considering the hadronization of $q\bar{q}$ strings of fixed kinematics, two simplifications are possible when evaluating the expression for $w_{\text{infer}}(e_h)$ of eq. (10). First, $p_{\text{sim}}^{\text{acc}}$ depends only on the initial string kinematics and endpoint quark flavors; if these are fixed, $p_{\text{sim}}^{\text{acc}}$ is just a number that can be calculated once from the baseline simulation. Second, it was found that with adequate data the average of eq. (10) need not be performed explicitly; instead, the substitution $\big\langle w_{\text{infer}}(\vec{S}_{jN_j}) \big\rangle_{e_j = e_h} \rightarrow w_{\text{infer}}(\vec{S}_{hN_h})$ can be made, finding it sufficient to consider each event separately.

The logarithm of the string break weights is parameterized as a difference of two neural networks (NNs) $g_1$ and $g_2$,

$$\ln w_s^{\text{infer}}(\vec{s}, \theta) = \ln g_\theta(\vec{s}) = g_1(z, \Delta\vec{p}_{\text{T}}, m, \texttt{fromPos}, \vec{p}_{\text{T}}^{\text{string}}; \theta) - g_2(\vec{p}_{\text{T}}^{\text{string}}; \theta). \tag{12}$$

The weights for string fragmentation chains thus become functions of NN parameters $\theta$,

$$w_{\text{infer}}(\vec{S}_{hN_h}, \theta) = \prod_{b=1}^{N_{\text{had}}-2} w_s^{\text{infer}}(\vec{s}_{hN_h b}, \theta). \tag{13}$$

The values of $\theta$ are optimized by minimizing the difference between $w_{\text{infer}}(e_h, \theta)$ and $w_{\text{class}}(e_h)$. In ref. [3], this was achieved by minimizing the loss function that was a sum of the binary cross-entropy loss, encoding the difference between $w_{\text{infer}}$ and $w_{\text{class}}$, and a regularization term for the $g_{1,2}$ NNs. The explicit forms of the loss functions can be found in ref. [3].

Once the new fragmentation function is inferred in Step 2, HOMER can be used to reweight events generated from the baseline PYTHIA simulation. Step 3 of the HOMER method uses the learned weights for each individual string fragmentation $w_s^{\text{infer}}(\vec{s}_{hcb}, \theta)$ to reweight any new baseline PYTHIA simulation history by its HOMER weight

$$w_{\text{HOMER}}(\vec{\mathbf{S}}_h) = \prod_{c=1}^{N_h} \prod_{b=1}^{N_{hc}} w_s^{\text{infer}}(\vec{s}_{hcb}, \theta), \tag{14}$$

where $N_{hc}$ are the string breaks contained in chain $c$. Note that the above product of the weights is over all the string fragmentations in the PYTHIA simulation history, including the rejected string fragmentations. This is because in Step 3 the marginalization over simulation histories is not done explicitly as in Step 2 (yielding eq. (10)), but implicitly when computing any expectation value over events weighted by eq. (14). As detailed in ref. [3], averaging over simulation histories $\vec{\mathbf{S}}_h$ that result in the same event $e_h$ gives

$$w_{\text{class}}(e_h) \simeq w_{\text{infer}}(e_h) \simeq w_{\text{HOMER}}(e_h) \equiv \langle w_{\text{HOMER}}(\vec{\mathbf{S}}_j) \rangle_{e_j = e_h}. \tag{15}$$

Note that the `finalTwo` efficiencies are taken into account automatically, unlike in eq. (10). That is, the baseline PYTHIA simulation can be reweighted in a straightforward fashion to best match the data by simply assigning the weight $w_{\text{HOMER}}(\vec{\mathbf{S}}_h)$ of eq. (14) to each simulated event. The individual weights $w_s^{\text{infer}}(\vec{s}_{hcb}, \theta)$ can then be reinterpreted in terms of the transformed fragmentation function $f_{\text{infer}}(z)$, as shown in ref. [3].

## 3.2 Modifications to account for gluons

Next, consider the hadronization of strings with varying numbers of gluons. To make the analysis even more realistic, we allow the strings to have varying kinematics, as is the case for strings obtained at the final stage of a parton shower. Two serious complications arise that need to be addressed. First, the information gap between event-level and fragmentation-level information is increased. That is, the relation between $f(z)$ and the measurable quantities is more complicated for strings with gluons, even if the kinematics of the string and the number of gluons are held fixed. Second, introducing a variable number of gluons and kinks on the string, with varying kinematics, means that the treatment of the `finalTwo` filter must be reevaluated. Step 2 of the HOMER method must be modified to handle these complications.

### 3.2.1 Increase in information gap and decrease in effective statistics

First, we consider the case of a string of fixed kinematics with a fixed number of gluons. The presence of multiple string regions, defined by the gluon kinks, leads to a more complex relationship between fragmentation-level variables, mainly the individual $z$ values per emission, and the event-level variables, the hadron four-momenta. This has practical consequences; as the event-level variables become less predictive, the mean-squared-error between $w_{\text{class}}$ and $w_{\text{HOMER}}$ increases.[5]

---

[5]We illustrate this in appendix A using a toy model, where we quantify the increase in the information gap by comparing the Area-Under-the-Curve (AUC) of the classifiers based on $w_{\text{class}}$ and $w_{\text{HOMER}}$.

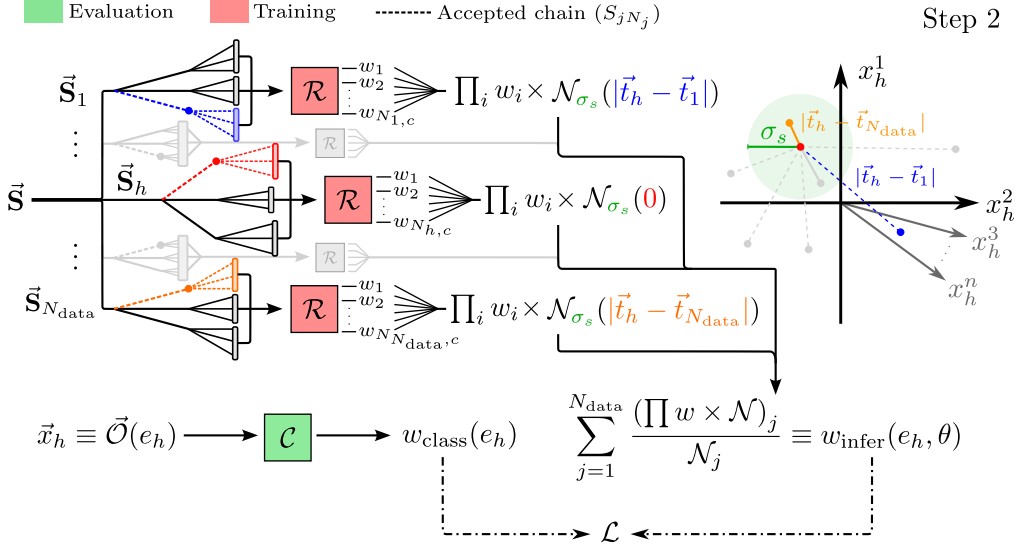

Figure 3: Flowchart of the modified Step 2 for the HOMER method with gluons.

To address this problem, we approximate the expectation value in eq. (10) with an average over a neighborhood of fragmentation chains,

$$\left\langle w_{\text{infer}}(\vec{S}_{jN_j}) \right\rangle_{e_j = e_h} \approx \frac{\sum_{\vec{S}_{jN_j}} w_{\text{infer}}(\vec{S}_{jN_j}) \mathcal{N}_{\sigma_s}\left(\left|\vec{t}_h - \vec{t}_j\right|\right)}{\sum_{\vec{S}_{jN_j}} \mathcal{N}_{\sigma_s}\left(\left|\vec{t}_h - \vec{t}_j\right|\right)} , \tag{16}$$

where $\mathcal{N}_{\sigma_s}$ is a Gaussian function of width $\sigma_s$ and $\vec{t}_h$ is a scaled vector of high-level observables $\vec{x}_h$ for event $e_h$

$$\vec{t}_h = \frac{\vec{x}_h - \min[\{\vec{x}_h\}]}{\max[\{\vec{x}_h\}] - \min[\{\vec{x}_h\}]} , \tag{17}$$

such that each component of $\vec{t}$ is between 0 and 1.[6] The similarity between two events $e_1$ and $e_2$ is thus measured by the distance $\left|\vec{t}_1 - \vec{t}_2\right|$, and the averaging of eq. (16) is performed over events that are within $\sigma_s$ of $e_h$ using this metric. This introduces some bias in the estimate of $\left\langle w_{\text{infer}}(\vec{S}_{jN_j}) \right\rangle_{e_j = e_h}$, which we find to be negligible as long as $\sigma_s$ is sufficiently small.

With this approximation, the Step 2 estimate of $w_{\text{infer}}(e_h, \theta)$ of eq. (10) is replaced by a smearing procedure

$$w_{\text{infer}}(e_h, \theta) = \frac{p_{\text{sim}}^{\text{acc}}}{p_{\text{data}}^{\text{acc}}(\theta)} \frac{\sum_{\vec{S}_{jN_j}} w_{\text{infer}}(\vec{S}_{jN_j}, \theta) \mathcal{N}_{\sigma_s}\left(\left|\vec{t}_h - \vec{t}_j\right|\right)}{\sum_{\vec{S}_{jN_j}} \mathcal{N}_{\sigma_s}\left(\left|\vec{t}_h - \vec{t}_j\right|\right)} , \tag{18}$$

while the rest of Step 2 remains unchanged. Here, $w_{\text{infer}}(\vec{S}_{hN_h}, \theta)$ is still given by a product of single emission weights from eq. (13).

### 3.2.2 Varying initial states

Strings of varying kinematics and with varying numbers of gluons introduce new challenges. Since the `finalTwo` efficiency depends on these initial state quantities, it is no longer practical

---

[6]Here, $\min[\{\vec{x}_h\}]$ denotes a vector where each component is the minimum of each high-level observable contained in $\vec{x}_h$, and $\max[\{\vec{x}_h\}]$ a vector containing each maximum. The high-level observables that we use are listed in section 4.

to calculate $p_{\text{sim}}^{\text{acc}}$ in eq. (18) once for each possible initial state. Instead, the smearing procedure is modified to estimate also the `finalTwo` efficiency by averaging over complete simulated histories, including the rejected chains

$$w_{\text{infer}}(e_h, \theta) = \frac{\sum_j w_{\text{HOMER}}(\vec{\mathbf{S}}_j) \mathcal{N}_{\sigma_s}(|\vec{t}_h - \vec{t}_j|)}{\sum_j \mathcal{N}_{\sigma_s}(|\vec{t}_h - \vec{t}_j|)}, \tag{19}$$

where the summation is over simulated histories. Specifically, in eq. (18) $w_{\text{infer}}$ is only calculated using the final accepted fragmentation chain $\vec{S}_{jN_j}$, while here all the fragmentation chains of the history $\vec{\mathbf{S}}_j$ are used. The factor $\mathcal{N}_{\sigma_s}(|\vec{t}_h - \vec{t}_j|)$ constrains the possible contributions from accepted fragmentation chains; only those that result in high-level observables similar to the one for the event $e_h$ contribute. The rejected fragmentation chains are not constrained at all by the $\mathcal{N}_{\sigma_s}$ factor, so that the summation over $j$ results in a good estimate of the effect of the `finalTwo` filter, at the cost of a decreased effective sample size due to the increased variance of the weights, $\sum w_{\text{HOMER}}^2$.

Equation (19) is a main result of this work; it is the required modification of Step 2 in HOMER to allow strings with gluons and of varying kinematics. With this modification, displayed in fig. 3, the HOMER weights are now utilized for both Step 2 and Step 3, first with smearing for training and then without smearing for reweighting any generated events with a fixed data-driven fragmentation function. We emphasize that while Step 2 of HOMER $w_{\text{infer}}(e_h)$ is now calculated using eq. (19) rather than eq. (10), all the other parts of HOMER remain the same. In particular, we use the same loss functions and neural network architectures as in ref. [3].

Apart from reduction of the effective sample size, the use of a smearing kernel also increases the numerical costs, since it requires a computation of all pairwise distances in a batch. In this work we found the increase in the numerical costs still manageable, though it may also be improved with more efficient computation and storage of distance matrices. Alternatives to the use of a smearing kernel do exist, but in general require further modeling. The smearing kernel, although numerically expensive, thus may well provide the simplest approach to averaging.

### 3.2.3 Selecting the hyperparameter $\sigma_s$

The width $\sigma_s$ of the smearing kernel in eq. (19) is a hyperparameter that can be optimized such that the weights $w_{\text{HOMER}}(\vec{\mathbf{S}}_h)$ lead to the best description of data. For very large $\sigma_s$, $\sigma_s \to \infty$, the inferred weights converge as $w_{\text{infer}}(e_h) \to 1$, *i.e.*, $w_{\text{infer}}$ becomes a random classifier with no information about the event. In the opposite limit, $\sigma_s \to 0$, the smearing reverts to the original HOMER approximation, see the discussion above for eq. (12), albeit now with a very poor approximation for the effect of the `finalTwo` filter. We thus expect that there is a small but nonzero value of $\sigma_s$ that provides an optimal choice.

To find the optimal value of $\sigma_s$, we use a $\chi^2$-inspired goodness of fit metric

$$\frac{\chi^2(\mathcal{O}, \sigma_s)}{N_{\text{bins}}} = \frac{1}{N_{\text{bins}}} \sum_{k=1}^{N_{\text{bins}}} \frac{\left(p_{\text{data},k}^{\mathcal{O}} - p_{\text{pred},k}^{\mathcal{O}}(\sigma_s)\right)^2}{(\sigma_{\text{data},k}^{\mathcal{O}})^2 + (\sigma_{\text{pred},k}^{\mathcal{O}}(\sigma_s))^2}, \tag{20}$$

where we choose the observable $\mathcal{O}$ to be the output of the classifier in Step 1, *i.e.*, $\mathcal{O} = -2\ln w_{\text{class}}$ in analogy to the test statistic used for hypothesis tests [15]. The value of $\mathcal{O}$ for each event is used to separate data into $N_{\text{bins}}$ bins, where $p_{\text{data},k}^{\mathcal{O}}$ in eq. (20) denotes the fraction of experimental events in bin $k$, and $p_{\text{pred},k}^{\mathcal{O}}(\sigma_s)$ is the corresponding predicted fraction obtained using HOMER weights $w_{\text{HOMER}}$. The $p_{\text{pred},k}^{\mathcal{O}}$ fractions depend on the hyperparameter $\sigma_s$

since $w_{\text{HOMER}}$ weights change depending on what value of $\sigma_{\text{s}}$ is used in the smearing of eq. (19). Both the measurement $\sigma^{\mathcal{O}}_{\text{data},k}$ and the simulation uncertainties $\sigma^{\mathcal{O}}_{\text{pred},k}(\sigma_{\text{s}})$ are used in the definition of the goodness-of-fit metric to adequately account for the impact of low statistics in some of the bins.

Minimization of $\chi^2(\mathcal{O},\sigma_{\text{s}})/N_{\text{bins}}$ gives the optimal value of the smearing hyperparameter $\sigma^*_{\text{s}}$. This optimization procedure may be numerically expensive since one needs to determine HOMER weights for several $\sigma_{\text{s}}$ values. In practice, the procedure can be accelerated via parallelization and a judicious exploration of possible $\sigma_{\text{s}}$ values. Note that the value of $\sigma^*_{\text{s}}$ depends on the size of data samples used to extract $w_{\text{HOMER}}$; for larger samples, $\sigma_{\text{s}}$ can be smaller and still result in sufficiently efficient smearing. This is particularly important since, due to memory constraints, we always compute the smearing on relatively small batches of $10^4$ events. We leave for future work a systematic exploration of the impact of sample and batch size on $\sigma^*_{\text{s}}$. However, we expect the numerical results shown in the next section to be further improved if larger computing resources were devoted to increasing both sample and batch sizes.

## 4  Numerical results

In our numerical studies here, we consider illustrative datasets corresponding to four different initial state scenarios, ordered by increasing complexity.

1. **Fixed $q\bar{q}$**: the $q\bar{q}$ string scenario of ref. [3], to highlight the challenges faced in the cases of hadronizations of strings with gluons.

2. **Fixed $qg\bar{q}$**: a $qg\bar{q}$ string with a fixed kinematic configuration determined by the three four-vectors $p_q$ $p_g$, and $p_{\bar{q}}$ of the initial quark, gluon, and anti-quark, respectively. See section 4.4.1 for more details.

3. **Variable $qg\bar{q}$**: $qg\bar{q}$ strings of differing string kinematics. See section 4.4.2 for details.

4. **Variable $qg^{(n)}\bar{q}$**: $q\bar{q}$ strings with an arbitrary number of $n$ gluons attached and with varying kinematics, as obtained from a parton shower. See section 4.4.3 for details.

In section 4.3, we show the results of $\sigma_{\text{s}}$ optimization for all three scenarios with gluons, and validate against the results for the $q\bar{q}$ string scenario.[7] Sections 4.4.1 to 4.4.3 then contain the discussion of HOMER reweighted predictions for the optimal choices of $\sigma_{\text{s}}$.

### 4.1  Numerical simulation details

Many of our choices in the numerical simulations follow the ones made in ref. [3]. For instance, for each of the initial state scenarios, we used PYTHIA to generate two different sets of $2 \times 10^6$ events, a baseline simulation dataset and a synthetic experimental dataset; for simplicity in both datasets, we only allowed the production of pions during hadronization. The baseline simulation datasets were generated using the default Monashref. [16] values for the parameters, $a_{\text{sim}} = 0.68$, $b = 0.98$, and $\sigma_{\text{T}} = 0.335$, while the synthetic experimental data was generated with a modified value of $a_{\text{data}} = 0.30$ with all the other parameters kept the same. Since both the baseline and experimental datasets were generated using PYTHIA, a closure test can be performed to check if the learned and the true hadronization functions match. In future applications, only the baseline dataset will be generated using PYTHIA, while observables measured from actual experiments will be used for Step 1. Although the synthetic experimental data is sufficient to study the performance of the model and allows for easy comparison with

---

[7]The results for $q\bar{q}$ strings were obtained using the same simulated samples as in ref. [3].

ref. [3], for completeness we consider a slightly more sophisticated case in appendix C, where we vary simultaneously the $a$ and $b$ parameters.

The $2 \times 10^6$ datasets were split in half, with $N_{\text{train}} = 10^6$ and $N_{\text{test}} = 10^6$ events in each dataset used for training and testing, respectively. All the figures below were obtained using the testing datasets, which were also used to verify the absence of any significant over-fitting both in Step 1 and Step 2 of the HOMER method. Motivated by the results obtained in ref. [3], high-level observables provided on an event-by-event basis were used to train the classifier in Step 1. These high-level observables are the same 13 observables that were used in ref. [3] motivated by the Monash tune [16]: event-shape observables $1-T$, $B_T$, $B_W$, $C$, and $D$; particle multiplicity $n_f$ and charged particle multiplicity $n_{\text{ch}}$; and the first three moments of the $|\ln x|$ distribution for all visible particles $\ln x_f$ and for the charged particles $\ln x_{\text{ch}}$.

To train the Step 1 classifier, we minimized the same loss function as in ref. [3]. The classifier outputs then determine the $w_{\text{class}}$ event weights via eq. (8). For this task, we used a gradient boosting classifier (GBC) implemented in XGBOOST [17] since it provides a fast, simple and powerful classifier that is applicable to the high-level event-by-event representation of the datasets; and found that the previously chosen hyperparameters[8] work well for all three cases. This choice was manually validated by observing whether the resulting classifiers were smooth and well-calibrated.

The main change in Step 2 is how the weights $w_{\text{infer}}$ were obtained, for which we now used eq. (19), while the rest of Step 2 followed closely ref. [3]. In particular, the training method of Step 2 was the same as in ref. [3] with the seven variables used to describe string breaks, see eq. (3), partitioned into two groups: The first group $z$, $\Delta p_x$, $\Delta p_y$, $m$, and `fromPos` characterizes each string break, while the second group $\vec{p}_{\text{T}}^{\text{string}} = \{p_x^{\text{string}}, p_y^{\text{string}}\}$ encodes the state of the string fragment before the break.

To minimize the Step 2 loss function, which has the same functional form as in ref. [3], we used a message passing graph neural network (MPGNN) implemented in the PYTORCH GEOMETRIC library [18] since it allows us to efficiently treat each fragmentation chain as a variable-size particle cloud with no edges between the nodes[9] with string break vectors $\vec{s}_{hcb}$, $b = 1, \ldots, N_{h,c}$, as the nodes. The learnable function $\ln g_\theta = g_1 - g_2$ corresponds to an edge function that is evaluated on each node and produces updated weights for that node. The updated weight $w_{\text{infer}}(\vec{S}_{hN_h}, \theta)$ for the whole fragmentation chain is then obtained by summing $\ln g_\theta$ over all the nodes and exponentiating the sum, in comparison to eq. (13).

The $g_1$ and $g_2$ functions are fully connected neural networks with 3 layers of 64 neurons each and rectified-linear-unit (ReLU) activation functions. The inputs are string break vectors $\vec{s}_{hcb}$ for $g_1$ and $\vec{p}_{\text{T}}^{\text{string}}$ for $g_2$, see eq. (12). The output of each neural network is a real number with no activation function applied, and the weight for a single string break is given by $w_s^{\text{infer}} = \exp(g_1 - g_2)$, which is then combined to produce the event weight $w_{\text{infer}}(e_h)$. We minimized the loss function using the `Adam` optimizer with an initial learning rate of $10^{-3}$ that decreases by a factor of 10 if no improvement is found after 10 steps. We train for 150 epochs with batch sizes of $10^4$. We apply an early-stopping strategy with 20 step patience to avoid over-fitting.

## 4.2 Numerical support for smearing

Before showing the results of the $\sigma_s$ scans, let us first review the numerical results supporting the need for smearing. Table 1 shows the AUCs obtained using the Step 1 classifier weights

---

[8]These hyperparameters are a learning rate of 1, `max_depth:12`, and `min_child_weight:1000`. All the other parameters are set to their default values.

[9]We do not need to connect the string breaks since $\vec{s}_{hcb}$ already tracks the relevant information about the string state before the string break, $\vec{p}_{\text{T}}^{\text{string}}$, which enters the string break probability distributions, see eq. (1).

Table 1: AUC for the classifiers derived from $w_{\text{class}}$ (column 2) and $w_{\text{exact}}$ (column 3), as well as the ratio of the two AUCs (column 4) for the four different string scenarios.

| Scenario | AUC $w_{\text{class}}$ | AUC $w_{\text{exact}}$ | Ratio |
|---|---|---|---|
| Fixed $q\bar{q}$ | 0.691 | 0.762 | 0.907 |
| Fixed $qg\bar{q}$ | 0.717 | 0.811 | 0.884 |
| Variable $qg\bar{q}$ | 0.693 | 0.795 | 0.871 |
| Variable $qg^{(n)}\bar{q}$ | 0.610 | 0.838 | 0.728 |

Table 2: Ratio of effective to baseline simulation statistics, $n_{\text{eff}}/N_{\text{sim}}$, for $w_{\text{class}}$ and $w_{\text{exact}}$, and their ratio, for the four different string scenarios.

| Scenario | $n_{\text{eff}}/N_{\text{sim}}(w_{\text{class}})$ | $n_{\text{eff}}/N_{\text{sim}}(w_{\text{exact}})$ | Ratio |
|---|---|---|---|
| Fixed $q\bar{q}$ | 0.570 | 0.261 | 2.19 |
| Fixed $qg\bar{q}$ | 0.472 | 0.165 | 2.86 |
| Variable $qg\bar{q}$ | 0.544 | 0.183 | 2.98 |
| Variable $qg^{(n)}\bar{q}$ | 0.789 | 0.0572 | 13.8 |

$w_{\text{class}}$ (column 2). These should be compared with the AUCs for the exact weights $w_{\text{exact}}$ (column 3), representing the upper limit on HOMER's performance. While the weights $w_{\text{class}}$ only have access to event-level information, the weights $w_{\text{exact}}$ encode the full fragmentation chains, which include unobservable information. A significant drop from the $w_{\text{exact}}$ AUC to the $w_{\text{class}}$ AUC demonstrates an information gap between fragmentation-level and observable-level information. Given that the gap is due to unobservable information, it thus implies the need for averaging over fragmentation chains that lead to the same event, see eq. (10). Practically, we achieve this by smearing over event neighborhoods, see eq. (19). As the complexity of the datasets increases, so does the gap between the $w_{\text{exact}}$ AUC and the $w_{\text{class}}$ AUC. That is, the inclusion of the smearing in HOMER is expected to be essential for the $qg^{(n)}\bar{q}$ scenario but may not be for the $q\bar{q}$ scenario.

For the $qg^{(n)}\bar{q}$ scenario, the kinematics of the final-state hadrons are significantly determined by the parton shower process and not just by the process of hadronization. Since the parton shower is shared between the experimental and simulation datasets, there is less discriminatory power between the two datasets, resulting in a drop in the $w_{\text{class}}$ AUC. For the $w_{\text{exact}}$ AUC we observe the opposite, it increases for the $qg^{(n)}\bar{q}$ scenario compared to the other three scenarios. A likely reason is that in the $qg^{(n)}\bar{q}$ scenario, more hadrons are produced per event, on average, which means that $f(z)$ is sampled more. Since the $w_{\text{exact}}$ weights have access to the fragmentation-level information, the parton shower does not obscure the differences between the synthetic experimental and baseline simulation datasets. More samples per event simply lead to additional factors of $f(z)_{\text{data}}/f(z)_{\text{sim}}$ and thus larger discriminatory power between the synthetic experimental and baseline simulation datasets. However, this is entirely due to the unobservable information.

The performance of HOMER also depends on the effective statistics, defined as $n_{\text{eff}} = \left(\sum w\right)^2/\sum w^2$, with the sum running over the events of the sample.[10] The ratios between $n_{\text{eff}}$ and the baseline simulation dataset size $N_{\text{sim}}$ are shown in table 2. Since the variance of $w_{\text{exact}}$ is larger than for $w_{\text{class}}$, which is only sensitive to event-level information and thus maps

---

[10]For further details, see also the discussion for the toy model in appendix A.

fragmentation chains with different probabilities onto identical weights, the effective dataset sizes are always smaller for $w_{\text{exact}}$. Furthermore, there is a relatively modest drop in the effective $w_{\text{exact}}$ dataset sizes between the fixed $q\bar{q}$, fixed $qg\bar{q}$, and variable $qg\bar{q}$ scenarios, and a significant drop for the $qg^{(n)}\bar{q}$ scenario. This demonstrates that low effective statistics in the $qg^{(n)}\bar{q}$ scenario limits the efficacy of smearing.

## 4.3  Results of the $\sigma_{\text{s}}$ scans

To find the optimal $\sigma_{\text{s}}$ values for each of the four hadronizing string cases $\sigma_{\text{s}}^*$, we calculate the goodness-of-fit metric $\chi^2(\mathcal{O}, \sigma_{\text{s}})/N_{\text{bins}}$ of eq. (20) for several different $\sigma_{\text{s}}$ values. Here, we use $N_{\text{bins}} = 50$. The results are collected in fig. 4 and table 3. We find $\sigma_{\text{s}}^* = 0.045, 0.065, 0.050,$ and 0.065 for the fixed $q\bar{q}$, fixed $qg\bar{q}$, variable $qg\bar{q}$, and variable $qg^{(n)}\bar{q}$ scenarios, respectively. That is, for our setup $\sigma_{\text{s}}^* \sim 0.05$, although the exact values depend on the string scenario, sample size, and batch size.

In general, we find that values of $\sigma_{\text{s}}$ below $\sigma_{\text{s}}^*$ lead to a worse Step 2 performance, with larger differences between $w_{\text{infer}}$ and $w_{\text{class}}$, and do not appear to find the correct fragmentation function. For values of $\sigma_{\text{s}}$ somewhat above $\sigma_{\text{s}}^*$, a better Step 2 performance is obtained but at the expense of more prominent differences between the inferred and the actual fragmentation function evidenced by larger differences between $w_{\text{exact}}$ and $w_{\text{HOMER}}$. This demonstrates an over-fitting of the predicted $w_{\text{infer}}$ weights to the Step 1 classifier outputs, $w_{\text{class}}$. The best $\sigma_{\text{s}}$ represents a compromise between these two tendencies

From the improvements in goodness-of-fit of table 3, where the ratio between the values of $\chi^2(\mathcal{O}, \sigma_{\text{s}})/N_{\text{bins}}$ for the case of no smearing and the optimal smearing are compared, we observe how smearing also improves the goodness-of-fit metric for the fixed $q\bar{q}$ string scenario, although the improvement is more modest than for the three other scenarios. Furthermore, the value of $\sigma_{\text{s}}^*$ is smaller than for the three scenarios where gluons are added to the initial $q\bar{q}$ string.

In general, the improvements in goodness-of-fit from no smearing to optimal smearing reflect the underlying information gaps between hadron-level observables captured in $w_{\text{class}}$ and fragmentation-level information captured by $w_{\text{exact}}$, as well as the available effective statistics in each scenario, see tables 1 and 2. As the ratio between the $w_{\text{exact}}$ AUC and the $w_{\text{class}}$ AUC increases, the model's performance with no smearing worsens, demonstrating an increased information gap. Thus the effect of smearing becomes more critical. However, since $w_{\text{HOMER}}$ approximates $w_{\text{exact}}$, as the effective statistics $n_{\text{eff}}$ for $w_{\text{exact}}$ decreases, see table 2, the performance of the best model as measured by goodness-of-fit also worsens due to the increasingly limited statistical power of the dataset.

Table 3: Goodness-of-fit values, $\chi^2(\mathcal{O}, \sigma_{\text{s}})/N_{\text{bins}}$ of eq. (20), for the case where no smearing was used (column 2), compared to the case with an optimized $\sigma_{\text{s}}^*$ (column 3), as well as their ratio (column 4), for the four different string scenarios. The best $\sigma_{\text{s}}^*$ is displayed in parentheses in column 3.

| Scenario | No smearing | Best ($\sigma_{\text{s}}^*$) | Ratio |
|---|---|---|---|
| Fixed $q\bar{q}$ | 9.80 | 1.58 (0.045) | 6.20 |
| Fixed $qg\bar{q}$ | 183 | 2.25 (0.065) | 81.4 |
| Variable $qg\bar{q}$ | 211 | 3.22 (0.050) | 65.4 |
| Variable $qg^{(n)}\bar{q}$ | 594 | 13.2 (0.065) | 45.0 |

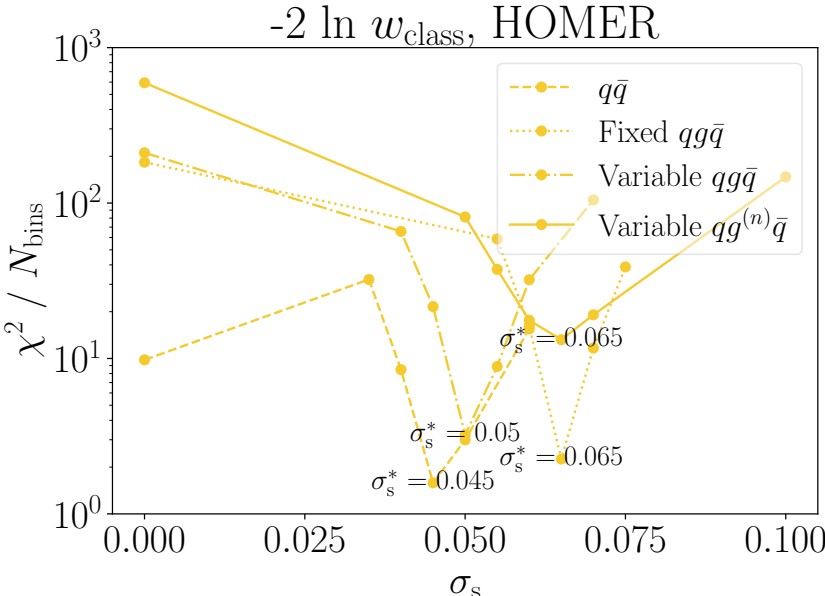

Figure 4: Goodness-of-fit defined by eq. (20), shown as a function of $\sigma_{\mathrm{s}}$ for the four different string scenarios with $N_{\mathrm{bins}} = 50$.

## 4.4 HOMER results for strings with gluons

Next, we examine the extraction of the fragmentation function $f(z)$ from synthetic data using the HOMER method for the three string scenarios including gluons: fixed $qg\bar{q}$ (section 4.4.1), variable $qg\bar{q}$ (section 4.4.2), and variable $qg^{(n)}\bar{q}$ (section 4.4.3). In each scenario the results are shown for the corresponding best value of the smearing hyperparameter $\sigma_{\mathrm{s}}^*$, obtained in section 4.3.

When plotting the results of the HOMER method, we use the following labels for different distributions.

- **Simulation**: the simulated distributions obtained using the baseline PYTHIA model.

- **Data**: the experimentally measured distributions. In our case, these are still synthetic data obtained using PYTHIA with the $a_{\mathrm{data}}$ Lund parameter instead of $a_{\mathrm{sim}}$, see section 4.1. One of the goals of the HOMER method is to reproduce these data distributions.

- **HOMER**: the results of the HOMER method, *i.e.*, distributions obtained by reweighting the simulation dataset with per event weights $w_{\mathrm{HOMER}}(e_h)$, eq. (14).

- **Exact weights**: the distributions obtained by reweighting the simulation dataset with the exact weights, following ref. [14]. These are the same weights that are used in section 4.2 to quantify the full fragmentation information. The *Exact weights* and *Data* distributions should match exactly, except for increased statistical uncertainties introduced by reweighting. The *Exact weights* distributions therefore represent an upper limit on the fidelity achievable by the HOMER method, due to the use of reweighting.

- **Best NN**: in this case the $g_1$ and $g_2$ NNs are trained directly on single emissions to learn the individual fragmentation function $f_{\mathrm{data}}(z)$. *Best NN* is a more realistic upper limit on the HOMER method than *Exact weights* since it also takes into account the limitations of approximating $f_{\mathrm{data}}(z)$ with $g_1$ and $g_2$.

In addition, we will show distributions that follow from intermediate stages of the HOMER method, with the following labels.

- **Classifier**: the distributions obtained by reweighting simulation datasets with per event weights $w_{\text{class}}(e_h)$, eq. (8), which were used in section 4.2 to quantify the high-level information. A comparison of the *Data* and *Classifier* distributions is a gauge of the performance of the classifier used in Step 1 of the HOMER method.

- **Inference**: similar to the classifier distributions, but using the per event weights $w_{\text{infer}}(e_h, \theta)$ obtained in Step 2. A comparison of the *Inference* and *HOMER* distributions measures the differences between two ways of performing the averages over the same event string fragmentation chains, eq. (10): either using smearing to calculate the event weights, eq. (19), or without the smearing, eq. (14), relying instead on the averaging being performed at the level of the event samples. In the limit of an infinite simulation sample sizes, these two averaging methods should match for the distributions of observables.

The agreement between the *Data* and all the other distributions for any observable $\mathcal{O}$ is quantified via the $\chi^2$ goodness-of-fit metric defined in eq. (20), where now $p^{\mathcal{O}}_{\text{pred},k}$ is the fraction of events in bin $k$ for the distribution of the observable $\mathcal{O}$, as predicted in any of the above cases: *Simulation*, *HOMER*, *Exact weights*, *Best NN*, *Classifier*, or *Inference*.

### 4.4.1 Fixed $qg\bar{q}$ scenario

We first consider the simplest case of a string with a gluon, the fixed $qg\bar{q}$ scenario where the initial state are $qg\bar{q}$ strings of a fixed kinematic configuration. As a benchmark point, we use a three jet configuration specified by the following three four-vectors $(E, p_x, p_y, p_z)$,

$$
\begin{aligned}
p_q &= (38.25, -6.75, 0.0, 37.65) \text{ GeV}, \\
p_g &= (13.5, 13.5, 0.0, 0.0) \text{ GeV}, \\
p_{\bar{q}} &= (38.25, -6.75, 0.0, -37.65) \text{ GeV},
\end{aligned}
\tag{21}
$$

where $p_q, p_g, p_{\bar{q}}$ are the four momenta of the initial quark, gluon, and anti-quark, respectively.

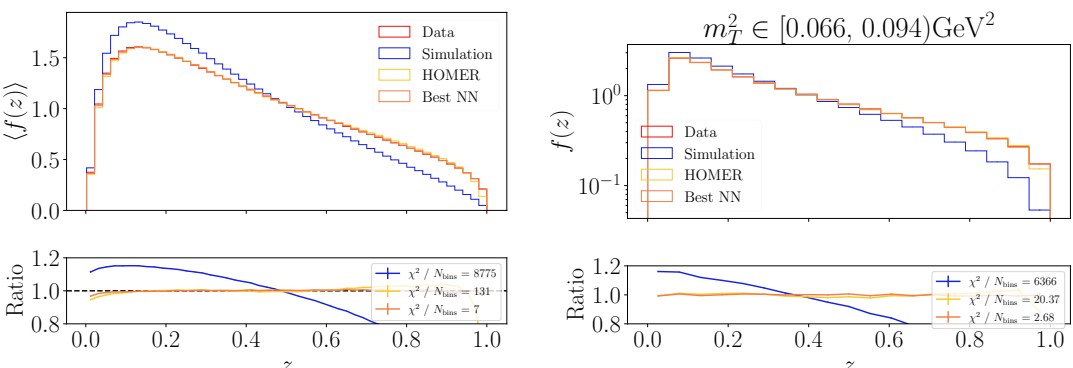

Figure 5: (left) Reweighted distributions for the fragmentation function averaged over all string break variables except $z$ and (right) fixing the transverse mass bin. All weights are from a model trained with the unbinned high-level observables for the fixed $qg\bar{q}$ scenario.

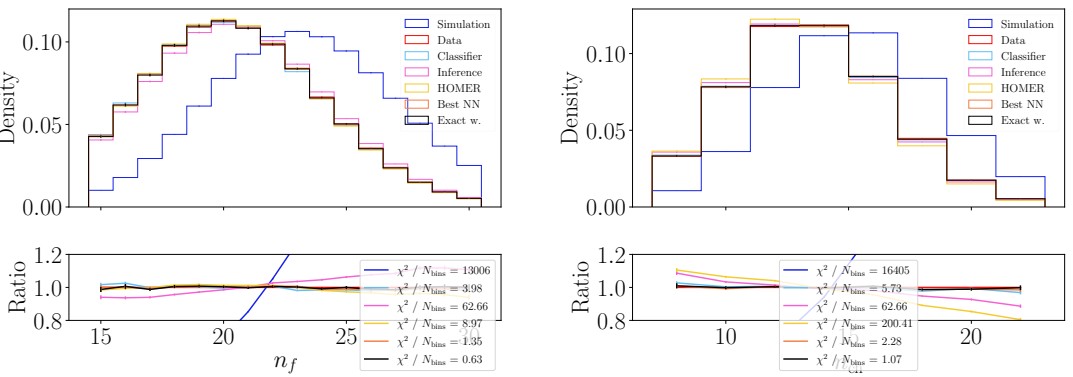

Figure 6: (left) Multiplicity and (right) charged multiplicity distributions for the fixed $qg\bar{q}$ scenario, where the training was performed on unbinned high-level observables.

Note that center of mass of the system is 90 GeV. The numerical results for the best value of the smearing hyperparameter $\sigma_s^* = 0.065$, are shown in figs. 5 to 7, with additional results collected in appendix B.1.

Figure 5 shows the Lund string fragmentation function that has been extracted using the HOMER method, *i.e.*, the form of $f(z)$ that is obtained by reweighting $f_{\text{sim}}(z)$ in each $z$ bin by the corresponding single hadron emission HOMER weights $w_s^{\text{infer}}(\vec{s}_{hcb}, \theta)$, see eq. (14). The right panel of fig. 5 shows $f(z)$ for a particular transverse mass bin, $m_T^2 \in [0.066, 0.094)$ GeV$^2$, while the left panel in fig. 5 gives the value of $f(z)$ averaged over all $m_T^2$ bins, and in both cases also averaged over all other variables. The HOMER extracted form of the fragmentation function agrees with $f_{\text{data}}(z)$ at the few percent level over most of the range of $z$. This is comparable, but somewhat worse, than the fidelity achieved in ref. [3] for the extraction of $f(z)$ from the hadronizations of the $q\bar{q}$ scenario. The degraded performance can be traced back to the intrinsic increase in degeneracies when going from fragmentation to high-level observables, *i.e.*, to the increased information gap between fragmentation-level and event-level observables that is encoded in the weights during the HOMER procedure.

The situation is similar for the HOMER predictions of hadron and charged hadron multiplicities, shown in left and right panels of fig. 6, respectively. The predictions are within a few percent of the *Data* distributions, where the fidelity for the hadron multiplicity is similar to the one obtained in the fixed $q\bar{q}$ scenario of ref. [3], and somewhat degraded for the charged hadron multiplicity. The hadron multiplicity is the most important feature used in the Step 1 classifier, according to the Shapley values [19,20], see fig. 7.[11] While charged multiplicity is, according to the Shapley values, the least important feature, this is likely due to a high correlation with the full multiplicity, so that $n_{\text{ch}}$ provides a similar classification performance as $n_f$. This situation is very similar to what was found for the fixed $q\bar{q}$ scenario, although the order of observables with subleading Shapley values did change. We also reiterate that the fidelity of the results depends on the value of the hyperparameter $\sigma_s$. The results shown were obtained with the optimized value $\sigma_s^*$, and the fidelity would degrade if sub-optimal values of $\sigma_s$ were used.

---

[11]Note that we use the SHAP implementation [19] for boosted decision trees that does not need to assume uncorrelated features. It relies on an interventional approach [21], where one studies the effect of fixing a particular feature on the model output. It is important to note, however, that Shapley values do not necessarily capture all the relevant information about the problem. For instance, the relationships between features are not captured by Shapley values, but can be studied through higher order interaction effects, see, e.g., [22].

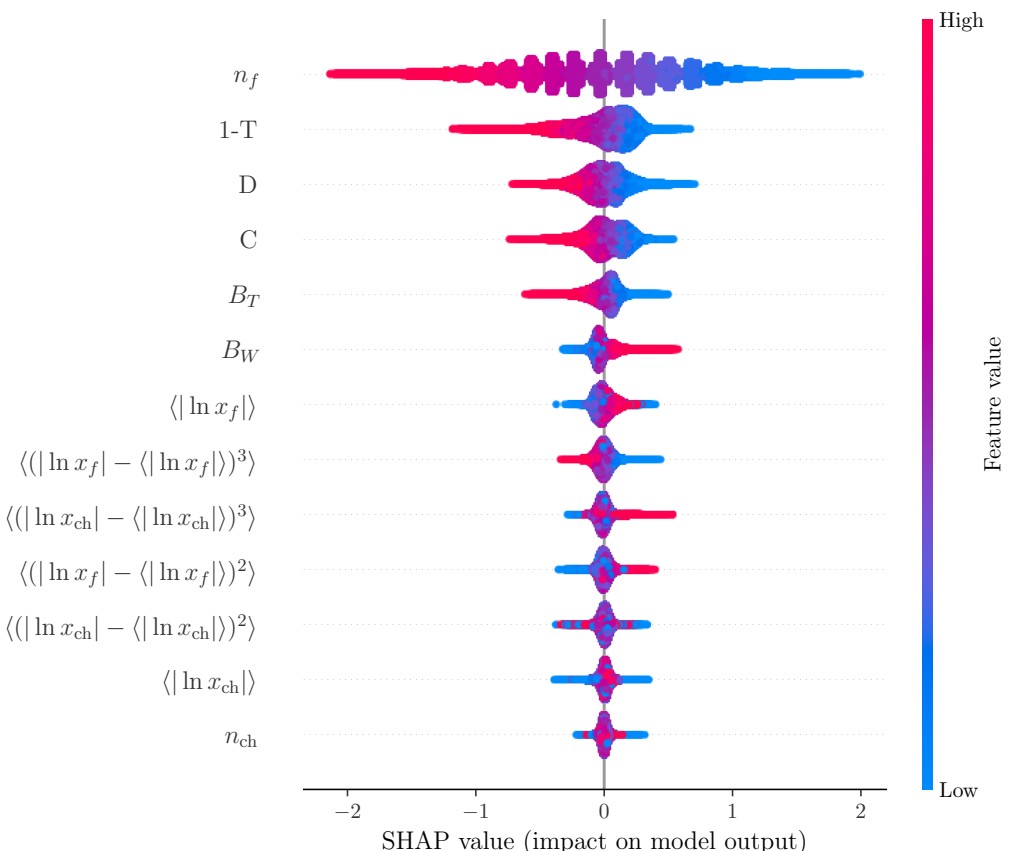

Figure 7: Shapley values for the classifier employed in Step 1 trained with the unbinned high-level observables for the fixed $qg\bar{q}$ scenario.

### 4.4.2 Variable $qg\bar{q}$ scenario

We now consider an ensemble of $qg\bar{q}$ strings with varying kinematics. The dataset is generated by starting from the $q\bar{q}$ initial state considered in ref. [3], *i.e.*, a $u\bar{u}$ pair in a color-singlet state with a center-of-mass energy $\sqrt{s} = 90$ GeV, and emitting a single gluon with the default `SimpleShower` implemented in PYTHIA before the system undergoes hadronization, where the emission can originate from either of the quarks. In this way, the resulting $qg\bar{q}$ system remains color-connected and is treated as a single string with varying initial state kinematics but a fixed center-of-mass energy. A comparison with the results of the fixed $qg\bar{q}$ scenario in section 4.4.1 highlights the effect of strings with different kinematic configurations on the extraction of the fragmentation function from data.

The main results, obtained with the optimal value for the smearing hyperparameter of $\sigma_s^* = 0.05$ are collected in figs. 8 and 9, with additional results relegated to appendix B.2. The extracted form of the fragmentation function $f(z)$, shown in fig. 8, agrees well with its functional form $f_{\text{data}}(z)$, roughly at the percent level, as it did in the fixed $qg\bar{q}$ string scenario of section 4.4.1. The fidelity of the extracted fragmentation function $f_{\text{HOMER}}(z)$, as measured by the goodness-of-fit metric remains comparable between fig. 8 and fig. 5.

Similar observations hold for the high-level observables, hadron multiplicity $n_f$ and charged hadron multiplicity $n_{\text{ch}}$, shown in the left and right panels of fig. 9, respectively. There is some improvement in fidelity of the $n_f$ and $n_{\text{ch}}$ distributions, as modeled by HOMER, com-

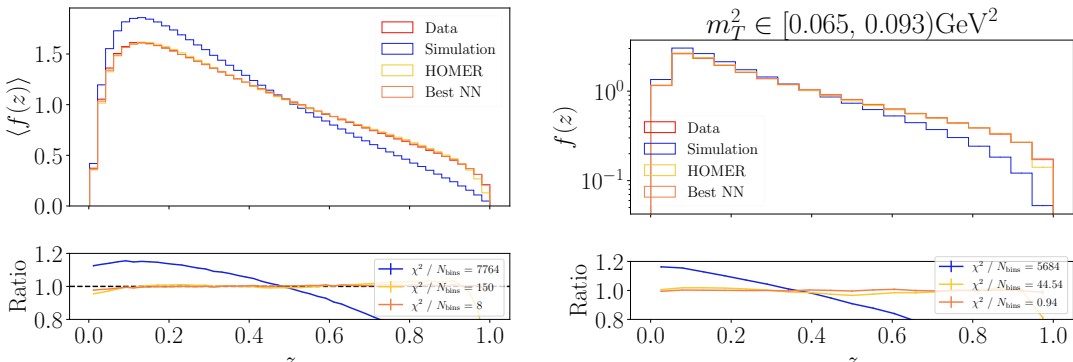

Figure 8: (left) Reweighted distributions for the fragmentation function averaged over all string break variables except $z$ and (right) fixing the transverse mass bin. All weights are from a model trained with the unbinned high-level observables for the variable $qg\bar{q}$ scenario.

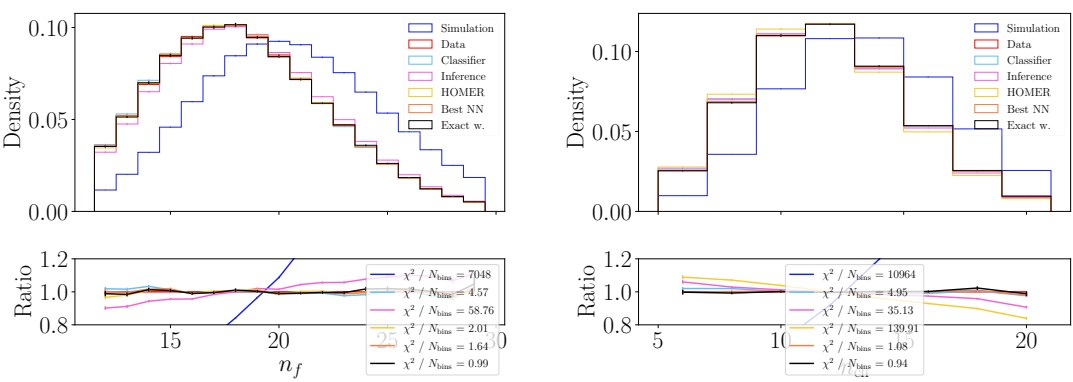

Figure 9: (left) Multiplicity and (right) charged multiplicity distributions for the variable $qg\bar{q}$ scenario, where the training was performed on unbinned high-level observables.

pared to the fixed $qg\bar{q}$ scenario, fig. 6. Interestingly, the HOMER prediction for $n_f$ is so close to the true distribution that it is of even higher fidelity than the distribution obtained in ref. [3] for the significantly simpler fixed $q\bar{q}$ scenario, where no smearing was performed in training. However, for the $n_{\text{ch}}$ distribution, the performance is marginally worse. These results illustrate how well the process of smearing in Step 2 of HOMER can overcome the added complexity introduced by gluons in the initial state.

### 4.4.3 Variable $qg^{(n)}\bar{q}$ scenario

Finally, we consider the most realistic scenario where we allow the initial $q\bar{q}$ color-singlet to undergo multiple gluon emissions via the default `SimpleShower` implemented in PYTHIA. Gluons originate from the initial quarks and subsequent emissions until the hadronization scale is reached. To keep the resulting system as one color-singlet composed of a $qg^{(n)}\bar{q}$ string, we only allow $q \to qg$ and $g \to gg$ splittings. Even with this approximation, and still limiting the analysis to a simplified flavor structure where only pions are produced, the obtained string dataset is already quite realistic, and can be representative of hadronic states produced by $e^+e^-$ collisions at a fixed center of mass energy. The results of a HOMER procedure, obtained for the optimized value of the smearing hyperparameter $\sigma_s^* = 0.065$ are shown in figs. 10 and 11, with additional results collected in appendix B.3.

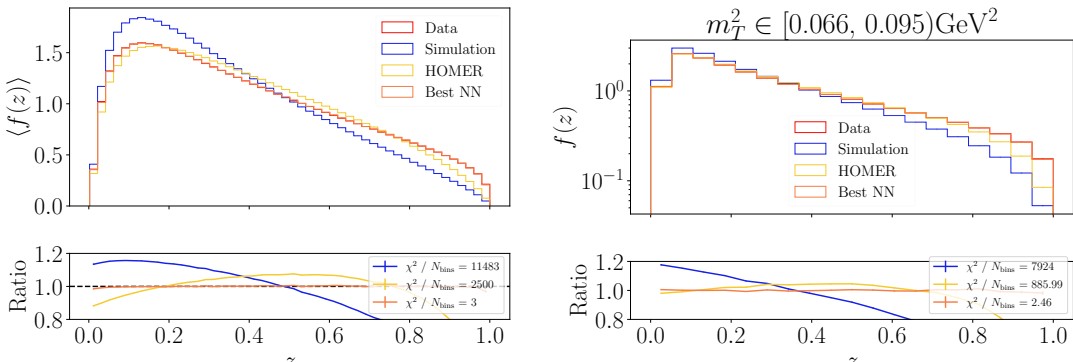

Figure 10: (left) Reweighted distributions for the fragmentation function averaged over all string break variables except $z$ and (right) fixing the transverse mass bin. All weights are from a model trained with the unbinned high-level observables for the variable $qg^{(n)}\bar{q}$ scenario.

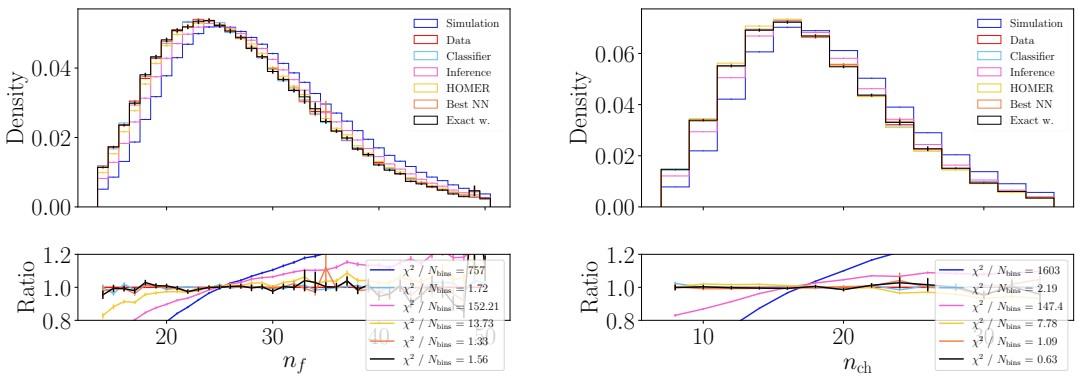

Figure 11: (left) Multiplicity and (right) charged multiplicity distributions for the variable $qg^{(n)}\bar{q}$ scenario, where the training was performed on unbinned high-level observables.

The extracted value of the fragmentation function $f(z)$, shown in fig. 10, shows deviations from its analytic form at the level of 5%, and is thus much larger than in the simpler fixed and variable $qg\bar{q}$ scenarios. This decrease in fidelity could have been anticipated already from the decreased performance of the classifier used in Step 1 of the HOMER method, which we encountered in section 4.2, see also table 1. This decrease in performance comes from two sources. First, the hadronization effects are now harder to factor out, since they are now combined with the effects of parton shower. Second, the increased variance in the exact weights implies a large decrease in effective statistics, making the training more challenging. The reduction in the classifier's quality achieved in Step 1 has downstream consequences, resulting in a decreased performance of the Step 2 output and the final HOMER predictions. However, this drop in fidelity is more pronounced for the extracted fragmentation function, fig. 10, than it is for the multiplicity observables shown in fig. 11.

Additional information regarding decreased performance of Step 1 classifier is provided in appendix B. For instance, fig. 26 in appendix B.3 shows the distribution of various weights (*Exact weights, Best NN, Classifier, Inference,* and *HOMER*) for the $qg^{(n)}\bar{q}$ scenario, as well as a correlation between the $w_{\text{infer}}$ and $w_{\text{class}}$ weights. The Pearson correlation coefficient $r$ between

$w_{\text{infer}}$ and $w_{\text{class}}$ is significantly lower than for the fixed and variable $qg\bar{q}$ scenarios, see fig. 18 and fig. 22, respectively. From this result one can conclude that a better choice of observables might increase the performance, where full phase space measurements, *i.e.*, the point cloud training dataset considered in ref. [3], warrant further exploration.

# 5 Conclusions and outlook

In this manuscript, we developed a framework that allows for an efficient solution of the inverse problem for hadronization, at least in its limited sense, the extraction of the Lund string fragmentation function $f(z)$ from data. The proposed version of the HOMER method is an extension of the method we introduced in ref. [3], where it was applied to the simpler case of $q\bar{q}$ strings with fixed kinematics. This simplified scenario made the introduction of the HOMER method more transparent and the results easier to interpret. The simplicity of a $q\bar{q}$ string also allowed for: (1) an easier interpretation of the performance of the classifier used in Step 1 of the HOMER method, where, in particular, there was no danger for parton showers to mask differences between hadronization models; and (2) the simplicity of the fixed initial state allowed for a straightforward inclusion of the efficiency factor due to fragmentation chains rejected during the simulation.

The main challenge in solving the inverse problem for hadronization is the fact that the mapping between the single hadron fragmentations and the observables that can be measured in an event is not bijective, *i.e.*, is not one-to-one. Several different fragmentation chains can result in the same event: the same configuration of hadrons, including their four momenta, up to some arbitrarily small numerical differences, and flavor compositions. In order to calculate the probability for a given event one thus needs to average over string fragmentation neighbors, *i.e.*, over string fragmentation chains that result in the same event. In this manuscript, we introduced a numerically efficient way to achieve this by performing smearing over these event neighbors, see eq. (19), though at the cost of some bias. This bias can be reduced, provided that the batch sizes over which the smearing is performed can be suitably increased.

As shown in section 4.4, the improved HOMER method can now be used to extract $f(z)$ in the more realistic case of strings containing multiple gluons, as would be obtained as the end result of a parton shower. In our numerical studies we considered three scenarios of increasing complexity. In section 4.4.1, we first considered the simplest case of $qg\bar{q}$ strings with a fixed kinematic configuration. In section 4.4.2, we considered an ensemble of $qg\bar{q}$ strings where the default PYTHIA parton shower determined the kinematics of the gluon. In section 4.4.3, we finally considered an ensemble of strings with an arbitrary number of gluons, whose kinematics were also given by the default PYTHIA parton shower. In all three cases, we demonstrated that the HOMER method allows for extracting the Lund string fragmentation function $f(z)$ without requiring a parametric form. While the fidelity of the extracted Lund fragmentation function $f(z)$ decreases with the complexity of the scenario, from the fixed $qg\bar{q}$ scenario to the variable $qg^{(n)}\bar{q}$ scenario, the achieved precision is expected to be still comparable with other experimental systematic uncertainties. Furthermore, we expect the fidelity to improve by simply scaling the overall sample and training batch sizes accordingly.

Thus far, we have only used synthetic data sets to train HOMER. The experimental data used in this proof-of-principle demonstration of the HOMER method was simulated with PYTHIA for a particular set of Lund string model parameter values. Using this synthetic data was essential to demonstrate a closure test, that extracting $f(z)$ with high enough fidelity is possible using only the observable information. Following the results shown here, we expect to extract $f(z)$ from experimentally measured distributions. However, several changes to the proof-of-principle analysis performed here will be needed.

1. Most of the experimental measurements currently available are only for binned high-level observables, see ref. [3], which is expected to degrade the fidelity of the extracted $f(z)$ [3].

2. When comparing to data, the simplified parton shower considered in this work, where only $q \to qg$ and $g \to gg$ splittings are allowed in the parton shower evolution, will need to be replaced with a full leading-log parton shower. Although using a more realistic parton shower will introduce additional complexity, including the possibility of multiple strings in a single event and other color topologies such as junctions, we expect HOMER to translate straightforwardly in implementation to this case. Further investigation will, however, be needed to gauge the impact on the performance of HOMER due to the possible degradation during Step 1.

3. Similarly, the simplifying assumption about the flavor structure of emitted hadrons, where we have restricted the hadronization to just pions, can be straightforwardly relaxed by relying on the PYTHIA flavor selector. While we expect only a limited impact on the performance of HOMER, this must still be demonstrated.

4. To offset any decrease in the performance of the Step 1 of HOMER, when comparing to data, we must identify a set of observables that are the most sensitive to hadronization. Ideally these measured observables should be available as unbinned, i.e., sets of observables on an event-by-event basis, which is the information that was used in training here.

5. All of these advancements should be supplemented by a robust uncertainty quantification. All numerical results shown in this work, including $\sigma_s^*$, correspond to a single HOMER run per dataset per hyperparameter. A more detailed investigation into the robustness of the method, including the assignment of uncertainties to the determination of $\sigma_s^*$ and to all learned weights, would be highly beneficial.

6. Finally, ideally, we could move to train on particle cloud-type event-by-event data once and if these measurements become available in the future.

While this proposed research program may appear daunting at first, steady progress can be made toward training HOMER on real data in the near future.

## Acknowledgments

We thank A. Ore, S. Palacios Schweitzer, T. Plehn and T. Sjöstrand for their careful reading and constructive comments on the manuscript.

**Funding information** AY, BA, JZ, MS, and TM acknowledge support in part by the DOE grant DE-SC1019775, and the NSF grants OAC-2103889, OAC-2411215, and OAC-2417682. JZ acknowledges support in part by the Miller Institute for Basic Research in Science, University of California Berkeley. SM is supported by the Fermi Research Alliance, LLC under Contract No. DE-AC02-07CH11359 with the U.S. Department of Energy, Office of Science, Office of High Energy Physics. CB acknowledges support from the Knut and Alice Wallenberg foundation, contract number 2017.0036. PI and MW are supported by NSF grants OAC-2103889, OAC-2411215, OAC-2417682, and NSF-PHY-2209769. TM acknowledges support in part by the U.S. Department of Energy, Office of Science, Office of Workforce Development for Teachers and Scientists, Office of Science Graduate Student Research (SCGSR) program. The SCGSR

program is administered by the Oak Ridge Institute for Science and Education for the DOE under contract number DE-SC0014664. This work is supported by the Visiting Scholars Award Program of the Universities Research Association. This work was performed in part at Aspen Center for Physics, which is supported by the NSF grant PHY-2210452.

## A   Smearing with a toy model

Here, we use a toy model to gain further insight about the smearing procedure and the reasons for why it is needed. For the toy example, let us consider a dataset of $N = 10^3$ events, where each event consists of a single number, $\hat{\mu}_n$, for $n = 1, \ldots, N$. The value of $\hat{\mu}_n$ is obtained by generating $L$ real numbers, $x_{nl}$, from a Gaussian distribution $\mathcal{N}$ with mean $\mu$ and standard deviation $\sigma$,

$$x_{nl} \sim \mathcal{N}(\mu, \sigma), \qquad n = 1, \ldots, N, \qquad l = 1, \ldots, L, \tag{A.1}$$

and then calculating the mean,

$$\hat{\mu}_n = \frac{1}{L} \sum_{l=1}^{L} x_{nl}. \tag{A.2}$$

This toy model has features that resemble the simulation of fragmentation as discussed in the main text. The simulation history in the toy model contains information, the values of $x_{nl}$, that is not observable in the event, since the only observables in the toy model are the values of the means, $\hat{\mu}_n$. This is similar to the fragmentation simulation histories, which contain additional information, the values of the $z$ variables[12] that are unobservable from the hadronic event, *i.e.*, from the list of hadrons in the accepted fragmentation chain.

We can now apply the HOMER method to this toy model. As in the main text, we will have the baseline simulation model, generated using a Gaussian distribution with mean $\mu_1$ and standard deviation $\sigma_1$, and the actual data distributions generated from Gaussian distributions with mean and standard deviation values of $\mu_2$ and $\sigma_2$, respectively. Unlike for the process of fragmentation, we can calculate for the toy example the $w_{\text{class}}$ and $w_{\text{exact}}$ weights analytically. The $w_{\text{class}}$ weight can be obtained by observing that the estimator for the mean follows a Gaussian distribution with a rescaled standard deviation,

$$\hat{\mu}_n \sim \mathcal{N}(\mu, \sigma/\sqrt{L}). \tag{A.3}$$

The optimal $w_{\text{class}}$ is therefore given by

$$w_{\text{class}}(\hat{\mu}_n) = \frac{\mathcal{N}(\hat{\mu}_n; \mu_2, \sigma_2/\sqrt{L})}{\mathcal{N}(\hat{\mu}_n; \mu_1, \sigma_1/\sqrt{L})}. \tag{A.4}$$

The exact weights $w_{\text{exact}}$ are the same as $w_{\text{HOMER}}$ weights and are given by,

$$w_{\text{HOMER}}(\{x_{nl}\}) = \prod_{l=1}^{L} \frac{\mathcal{N}(x_{nl}; \mu_2, \sigma_2)}{\mathcal{N}(x_{nl}; \mu_1, \sigma_1)}, \tag{A.5}$$

where $\{x_{nl}\}$ denotes the full simulation history for event $n$, *i.e.*, all the values of the drawn numbers, $x_{nl}$ for $l = 1, \ldots, L$ that then generate $\hat{\mu}_n$. The two weights, $w_{\text{class}}(\mu_n)$ and $w_{\text{HOMER}}(\{x_{nl}\})$ are the same for $L = 1$, where $\hat{\mu}_n = x_{n1}$. We show the $x_{nl}$ and $\hat{\mu}_n$ distributions in the upper panels of fig. 12 for the case of $(\mu_1, \sigma_1) = (1.1, 1.2)$, $(\mu_2, \sigma_2) = (0.9, 0.9)$, and $L = 15$. The lower panels in fig. 12 show the distributions of the corresponding weights $w_{\text{class}}(\hat{\mu}_n)$ and

---

[12]In the problem of hadronization there is additional information that is unobservable, namely the rejected fragmentation chains, which do not have an equivalent in this toy model.

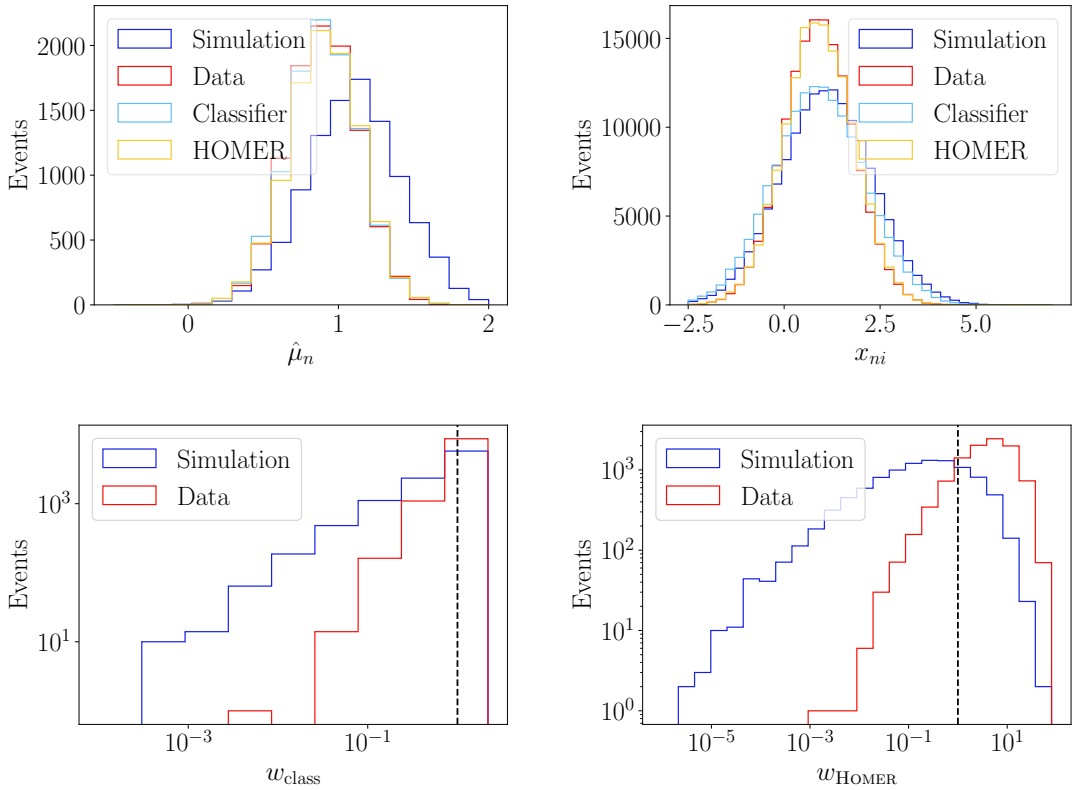

Figure 12: Toy model distributions for latent and observed variables, with the resulting classifier and exact weights. These plots are produced with $(\mu_1, \sigma_1) = (1.1, 1.2)$, $(\mu_2, \sigma_2) = (0.9, 0.9)$, $N = 10^3$ and $L = 15$.

$w_{\text{HOMER}}(\{x_{nl}\})$. Note, that for this choice of $\sigma_{1,2}$ and $\mu_{1,2}$ there is a considerable difference between the classifier weights $w_{\text{class}}$ and the exact weights, $w_{\text{HOMER}}$, with $w_{\text{HOMER}}$ a much more powerful discriminant between simulation and data.

Next, we illustrate with this toy model the need for introducing smearing in the HOMER method. In the main text we used HOMER to learn in a data-driven way the fragmentation function by matching the $w_{\text{HOMER}}$ and $w_{\text{class}}$ weights. The problem we are facing is a practical one; how we actually match these two sets of weights. For instance, in the toy model we can write a probabilistic model for $w_{\text{class}}(\hat{\mu}_n)$ by rewriting

$$\mathcal{N}(\hat{\mu}_n; \mu_k, \sigma_k/\sqrt{L}) = \int \delta\left(\hat{\mu}_n - \frac{1}{L}\sum_{l=1}^{L} x_l\right) \prod_{l=1}^{L} dx_l \, \mathcal{N}(x_l; \mu_k, \sigma_k). \tag{A.6}$$

Using this we can then write down the loss function for $w_{\text{HOMER}}$ as

$$\mathcal{L} = \frac{1}{N}\sum_{n=1}^{N}\left(w_{\text{class}}(\hat{\mu}_n) - \frac{\int \delta(\hat{\mu}_n - \frac{1}{L}\sum_{l=1}^{L} x_l) w_{\text{HOMER}}(\{x_l\}) \prod_{l=1}^{L} dx_l \, \mathcal{N}(x_l; \mu_1, \sigma_1)}{\int \delta(\hat{\mu}_n - \frac{1}{L}\sum_{l=1}^{L} x_l) \prod_{l=1}^{L} dx_l \, \mathcal{N}(x_l; \mu_1, \sigma_1)}\right)^2. \tag{A.7}$$

In practice, this loss function requires an inordinate amount of statistics, since we do not have a dedicated simulator that would sample $\{x_l\}$ conditioned on $\hat{\mu}_n$. The same problem is encountered in hadronization, where one would want to sample fragmentation histories conditioned on the observed hadronic event, which is very computationally demanding. In our previous work on the HOMER method applied to the hadronization of $q\bar{q}$ strings [3], we were able to circumvent this problem since the hadronizing system was significantly simpler.

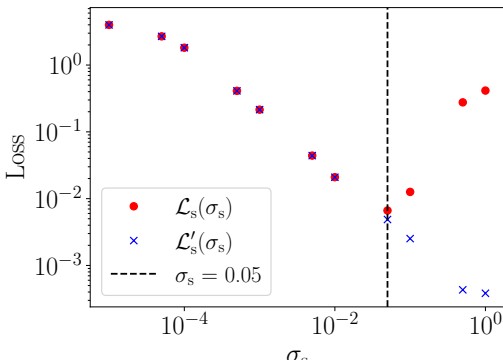

**Figure 13:** Loss function comparing $w_{\text{class}}$ and smeared $w_{\text{HOMER}}$ weights for different choices of $\sigma_s$, with the other parameters the same as in fig. 12.

Rather than minimizing the complicated loss function of eq. (A.7), we minimized the much simpler loss function[13]

$$
\mathcal{L}_0 = \frac{1}{N} \sum_{n=1}^{N} (w_{\text{class}}(\hat{\mu}_n) - w_{\text{HOMER}}(\{x_{nl}\}))^2 \,. \tag{A.8}
$$

That is, while the $w_{\text{HOMER}}(\{x_{nl}\})$ varies for different $\{x_{nl}\}$ sets, even if these give the same $\hat{\mu}_n$, the variation in $w_{\text{HOMER}}$ was still sufficiently small that the approximation $w_{\text{HOMER}}(\{x_{nl}\})$ *on average* equals $w_{\text{class}}(\hat{\mu}_n)$ was valid. In this work, we have seen that this approximation no longer holds when an additional gluon is added to the initial string. Instead, we need to consider the smeared loss function. In the toy model the smeared loss function is given by

$$
\mathcal{L}_s(\sigma_s) = \frac{1}{N} \sum_{n=1}^{N} (w_{\text{class}}(\hat{\mu}_n) - w_{\text{infer}}(\hat{\mu}_n, \sigma))^2 \,, \tag{A.9}
$$

where

$$
w_{\text{infer}}(\hat{\mu}_n, \sigma_s) = \frac{\sum_{m=1}^{N} w_{\text{HOMER}}(\{x_{ml}\}) \mathcal{N}(\hat{\mu}_m; \hat{\mu}_n, \sigma_s)}{\sum_{m=1}^{N} \mathcal{N}(\hat{\mu}_m; \hat{\mu}_n, \sigma_s)} \,, \tag{A.10}
$$

is the $w_{\text{HOMER}}$ weight averaged over neighbors in $\hat{\mu}$ space that are within roughly $\sigma_s$ distance around $\hat{\mu}_n$. In the main text these were the so-called inference weights $w_{\text{infer}}$; we thus use the same notation here. That is, instead of eq. (A.7) where the average of $w_{\text{HOMER}}(\{x_{nl}\})$ is over all the $\{x_{nl}\}$ that result in a given $\hat{\mu}_n$, we have replaced it in eq. (A.9) with an average over the neighbors. In the large $N$ limit we can reduce the size of the neighborhood to averages over, ultimately taking $\sigma_s \to 0$ when $L \to \infty$, *i.e.*, taking the limits in such a way that there is still a large number of neighbors to average over. That is, in the large $N$ limit we have $\mathcal{L}_s(\sigma_s = 0) = \mathcal{L}$.

The loss function $\mathcal{L}_s(\sigma_s)$ in eq. (A.9) is much easier to evaluate in practice than the exact loss function $\mathcal{L}$ of eq. (A.7) is. Namely, the loss function $\mathcal{L}_s(\sigma_s)$ only requires computing the weighted averages over generated events, with no special requirements on the event generation. Furthermore, $\mathcal{L}_s(\sigma_s)$ approximates well enough the exact $\mathcal{L}$, such that it can be used in practical applications, as long as $N$ is large enough, and $\sigma_s$ is chosen judiciously. That is, $\sigma_s$ should be large enough to average over a sufficient sample of $w_{\text{HOMER}}(\{x_{nl}\})$, such that $w_{\text{infer}}(\hat{\mu}_n, \sigma_s)$ is a reasonable estimator of $w_{\text{class}}(\hat{\mu}_n)$. At the same time $\sigma_s$ should be small

---

[13]We also needed to account for `finalTwo`, which is not necessary in this toy model.

enough to not average over events with very different $\hat{\mu}_n$. The best choice of $\sigma_s$ can even be quantified; for this toy model we suggest the choice of $\sigma_s$ that minimizes the value of $\mathcal{L}_s$.

An example of this procedure is shown in fig. 13, where the red points show $\mathcal{L}_s$ evaluated for several values of $\sigma_s$, while the blue crosses show $\mathcal{L}'_s$, which is the same as $\mathcal{L}_s(\sigma_s)$ in eq. (A.9) but with $w_{\text{class}}(\hat{\mu}_n)$ replaced with a smeared version

$$w_{\text{class}}(\hat{\mu}_n, \sigma_s) = \frac{\sum_{m=1}^N w_{\text{class}}(\hat{\mu}_m)\mathcal{N}(\hat{\mu}_m; \hat{\mu}_n, \sigma_s)}{\sum_{m=1}^N \mathcal{N}(\hat{\mu}_m; \hat{\mu}_n, \sigma_s)} \, . \tag{A.11}$$

From fig. 13 we see that $\mathcal{L}'_s(\sigma_s)$ equals $\mathcal{L}_s(\sigma_s)$ for very small values of $\sigma_s$. For large values of $\sigma_s$ the loss function $\mathcal{L}_s(\sigma_s)$ starts to grow since $w_{\text{infer}}(\hat{\mu}_n, \sigma_s)$ is obtained from smearing over very different events, and thus $w_{\text{infer}}(\hat{\mu}_n, \sigma_s)$ no longer is a good approximation to $w_{\text{class}}(\hat{\mu}_n)$. The loss function $\mathcal{L}'_s(\sigma_s)$, however, keeps falling with growing $\sigma_s$, since $w_{\text{infer}}(\hat{\mu}_n, \sigma_s)$ improves as an approximation of $w_{\text{class}}(\hat{\mu}_n, \sigma_s)$ the larger the sample of events smeared over.

The value of $\mathcal{L}'_s(\sigma_s)$ appears to saturate for very large values of $\sigma_s$, when both smeared weights become almost random fluctuations around 1. For $\sigma_s$ large enough, both of these weights tend to one, while $\mathcal{L}'_s$ tends to zero. We see that the value of $\sigma_s$ for which $\mathcal{L}_s(\sigma_s)$ is minimal also corresponds to roughly the range of values of $\sigma_s$ for which the two loss functions start to differ appreciably. For the example shown in fig. 13 this occurs for $\sigma_s \simeq 0.05$.

Figure 14 further illustrates that the above data-driven prescription for finding the optimal value of $\sigma_s$ is well motivated. Here the left panels show the distributions of $\hat{\mu}_n$ and the right panels the various weights for three different values of $\sigma_s$. In the upper row the value of $\sigma_s$ was set to an extremely small value, such that the smearing has no effect; the dashed lines are indistinguishable from the solid lines. The distributions of (gold solid line) $w_{\text{HOMER}}$ and (blue solid line) $w_{\text{class}}$ weights differ because $w_{\text{class}}$ only has access to the observable event, *i.e.*, the value of $\hat{\mu}_n$, while $w_{\text{HOMER}}$ accesses all values of $x_{nl}$ that are unobservable. The equivalent estimation for the problem of hadronization addressed in the main text is that $w_{\text{class}}$ only depends on the final-state hadron momenta, while $w_{\text{HOMER}}$ depends on the simulated and unobservable $z$ values.

We observe how wide-ranging values of $w_{\text{HOMER}}$ can correspond to the same observable event. The role of smearing is to average over these different values of $w_{\text{HOMER}}$ to give a good approximation of $w_{\text{class}}$. In the middle right panel of fig. 14 we see that smearing with a close to optimal $\sigma_s = 0.05$ gives a (gold dashed line) $w_{\text{infer}}(\hat{\mu}_n, \sigma_s)$ that approximates the (blue solid line) classifier weights $w_{\text{class}}$ very well. The smearing is over a sufficiently small neighborhood in $\hat{\mu}_n$ space that it does not sculpt the distributions of the observables, the dashed blue and blue solid lines still match very well in the middle left panel, nor those of the classifier weights where the solid and dashed blue lines are almost indistinguishable in the middle right panel. This is no longer the case for (bottom row panels) very large values of $\sigma_s$, where smearing sculpts both the distributions of observables and the classifier weights.

The toy model can also provide a justification for why sometimes smearing may not be necessary. To do so, we take the previous examples and scan over the possible values of $L$, with the results shown in fig. 15. The top left panel in fig. 15 shows the ratio of the AUC between two classifiers, the $w_{\text{class}}$ and $w_{\text{HOMER}}$ weights. We see that AUC from $w_{\text{class}}$ drops relative to the AUC from $w_{\text{HOMER}}$ as the $L$ increases. This is expected, since with growing $L$ the estimated mean $\hat{\mu}_n$ contains less information than the collection of Gaussian samples $\{x_{nl}\}$ from which $\hat{\mu}_n$ is calculated.

The larger $L$ is, the less information $\hat{\mu}_n$ contains about each individual Gaussian sample, $\{x_{nl}\}$. This in turns increases the gap in the discriminatory power contained in the exact weights, $w_{\text{HOMER}}$, compared to the classifier weights, $w_{\text{class}}$. This can be seen, for instance, from the ratio $\text{AUC}_{\text{class}}/\text{AUC}_{\text{HOMER}}$ plotted in the top left panel of fig. 15 for several values of $L$. Here, $\text{AUC}_{\text{class}}$ is the AUC for the classifier that uses $w_{\text{class}}$, while $\text{AUC}_{\text{HOMER}}$ is the AUC



Figure 14: (left) Observable and (right) weight distributions for three choices of $\sigma_s$, with the other parameters as in fig. 12.

obtained using the $w_{\text{HOMER}}$ weights. We observe that as $L$ increases, the ratio $\text{AUC}_{\text{class}}/\text{AUC}_{\text{HOMER}}$ decreases, indicating, as expected, a relatively weaker discriminatory power of $w_{\text{class}}$ relative to $w_{\text{HOMER}}$. Eventually, the ratio $\text{AUC}_{\text{class}}/\text{AUC}_{\text{HOMER}}$ reaches a minimum and then starts to gradually increase with $L$. This can be explained by observing that for sufficiently large $L$, any further increase in $L$ only modifies the tails of the exact weight $w_{\text{HOMER}}$ distribution, resulting in marginal AUC gains, whereas the resulting classifier weight $w_{\text{class}}$ distributions are more impacted due to the decrease in the variance of the mean estimator $\hat{\mu}_n$.

The increase in $L$ modifies the effective statistics of the example. To capture this effect we follow ref. [14] to define the effective sample size $n_{\text{eff}}$ as

$$n_{\text{eff}} = \frac{\left(\sum w\right)^2}{\sum w^2}, \tag{A.12}$$

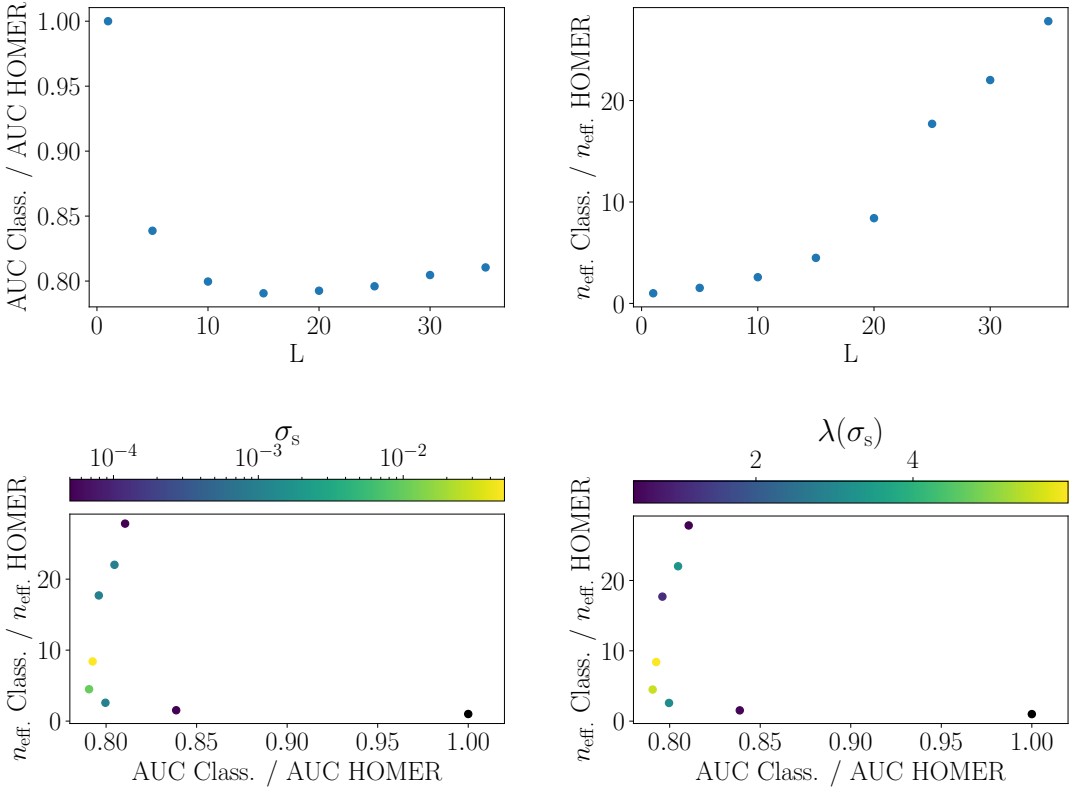

Figure 15: Results from scanning over possible choices of $L$, with the other parameters set as in fig. 12.

where the sum is over events, and $w$ is either $w_{\text{class}}$ or $w_{\text{HOMER}}$, giving $n_{\text{eff}}^{\text{class}}$ and $n_{\text{eff}}^{\text{HOMER}}$, respectively. We show the ratio $n_{\text{eff}}^{\text{class}}/n_{\text{eff}}^{\text{HOMER}}$ as a function of $L$ in the top right panel of fig. 15. We observe that, even though the performance of the classifier based on $w_{\text{HOMER}}$ increases both in absolute terms and relative to $w_{\text{class}}$ for large $L$, as shown in the top left panel in fig. 15, the exact weights $w_{\text{HOMER}}$ do suffer from very low statistics relative to the $w_{\text{class}}$ weights.

The interplay between performance and effective sample sizes determines the optimal value of $\sigma_s$. In the bottom row of fig. 15 we show in the $\text{AUC}_{\text{class}}/\text{AUC}_{\text{HOMER}}$ versus $n_{\text{eff}}^{\text{class}}/n_{\text{eff}}^{\text{HOMER}}$ plane the (left panel) best $\sigma_s$ and the (right panel) increase in the performance as measured by $\lambda(\sigma_s) = \ln\left[\mathcal{L}_0/\mathcal{L}_s(\sigma_s)\right]$. Here, $\mathcal{L}_0$ is the loss function that measures the difference between $w_{\text{class}}(\hat{\mu}_n)$ and $w_{\text{HOMER}}(\{x_{nl}\})$, see eq. (A.8), and thus contains inherent statistical fluctuations from $w_{\text{HOMER}}(\{x_{nl}\})$, while $\mathcal{L}_s(\sigma_s)$ measures the difference between $w_{\text{class}}$ and the smeared exact weights, $w_{\text{infer}}$, see eq. (A.9). If the smearing is adequate, then $w_{\text{infer}}$ would correctly approximate $w_{\text{class}}$, and thus $\mathcal{L}_s(\sigma_s) \ll \mathcal{L}_0$. In the right bottom panel of fig. 15 we see that this is indeed the case, with $\lambda(\sigma_s)$ taking values on the order of 1 for optimal $\sigma_s$ with the exact number depending on $L$. We also observe in the left panel of fig. 15 that the optimal value of $\sigma_s$ increases when the performance gap between $w_{\text{class}}$ and $w_{\text{HOMER}}$ as measured by the AUC is large, and that the optimal value of $\sigma_s$ decreases if the effective statistics for the $w_{\text{HOMER}}$-based classifier is much lower than for the $w_{\text{class}}$-based one. If the effective statistics are too low, then the $\sigma_s$ scan prefers smaller $\sigma_s$ values than the AUC ratio would suggest. That is, the minimization procedure is hindered by the effective statistics and settles for lower $\sigma_s$ values even at the expense of a smaller increase in performance as measured by $\lambda(\sigma_s)$.

This toy example exemplifies why we need to introduce smearing when dealing with more complicated string topologies. The relationship between fragmentation chains and observables becomes more complicated, with the former less determined by the latter. This results in an increased gap in weight performance. To compensate for this, we need to smear the fragmentation-based weights to better match the classifier weights. This is independent of the need to average out rejected chains if not accounting for `finalTwo` explicitly.

Figure 16: Distributions of high-level observables obtained at different stages of the HOMER method for the fixed $qg\bar{q}$ scenario. The corresponding multiplicity distributions are given in fig. 6.

# B  Additional results

In this appendix we collect additional figures that supplement the results shown in section 4. The additional figures give further information about the distributions of event weights, as well as about the fidelity of the HOMER predictions for other observables, beyond the multiplicities shown in the main text. Finally, we show the results for the optimal summary statistics, as defined in ref. [3].

## B.1  Fixed $qg\bar{q}$ scenario

In this subsection we collect additional figures to the ones shown in section 4.4.1 for the fixed $qg\bar{q}$ scenario.

Figure 16 shows the distributions for the high-level observables $1-T$, $C$, $D$, $B_W$, and $B_T$ for (blue) simulation, (red) data, and for the cases where the simulation datasets are reweighted using weights from different stages of the HOMER method, using (cyan) $w_{\text{class}}$, (magenta) $w_{\text{infer}}$, and (yellow) $w_{\text{HOMER}}$, while reweighted distributions obtained using (orange) *Exact weights* and (black) *Best NN weights* are also shown.[14] Similarly, fig. 17 shows the predictions for the moments of the $\ln x_f$ and $\ln x_{\text{ch}}$ distributions, where $x = 2|\vec{p}|/\sqrt{s}$ is the momentum fraction of a particle, such that $\sqrt{s}$ is the center-of-mass energy of the collision and $\vec{p}$ is the momentum of the particle. The moments are computed for each event both for all hadrons, $\ln x_f$, and just for charged hadrons, $\ln x_{\text{ch}}$. For all the observables the agreement between *Data* and the reweighted distributions is at the few percent level. This includes the multiplicity observables of fig. 6. Typically, the goodness-of-fit statistic $\chi^2/N_{\text{bins}}$, is comparable between the *Classifier* and *HOMER* distributions, and is somewhat higher for the *Inference* distributions. Most importantly, the difference between distributions obtained using the *Exact weights* and the *HOMER* predictions, is many orders of magnitude smaller than the difference between *Simulation* and *Data*.

Figure 18 shows the distributions of weights obtained during different stages of the HOMER method. Note that difference between (blue) *Classifier* and (black) *Exact weights* distributions is relatively modest, despite applying to different configurations. That is, $w_{\text{class}}(e_h)$ correspond to a ratio of probabilities for an event $e_h$ in simulation versus data, while $w_{\text{exact}}(\vec{\mathbf{S}}_h)$ is the ratio of probabilities for fragmentation histories. To go from $w_{\text{exact}}(\vec{\mathbf{S}}_h)$ to $w_{\text{class}}(e_h)$ one still needs to average over different fragmentation histories that lead to the same event $e_h$, compare to eq. (10). The relatively small difference between *Classifier* and *Exact weights* distributions in fig. 18 signals a relatively small information gap for the fixed $qg\bar{q}$ scenario, compared to the other string scenarios. The right panel of fig. 18 also shows a high correlation between $w_{\text{class}}$ and $w_{\text{infer}}$ weights, as also indicated by a relatively sizable Pearson correlation coefficient $r$. In the left panel of fig. 18 we also observe that the use of the optimal smearing results in a HOMER weights distribution that reproduces well the distribution of the $w_{\text{exact}}$ weights.

Figure 19 collects the predictions for the optimal summary statistic, $-2\ln w$. The left and right panels of fig. 19 show the distributions for $-2\ln w_{\text{class}}$ and $-2\ln w_{\text{exact}}$, respectively. While $-2\ln w_{\text{class}}$ can be calculated for each event without knowing the fragmentation histories the full simulation history is required for $-2\ln w_{\text{exact}}$. Consequently, this distribution can only calculated for data here because we are using synthetic data, *i.e.*, this is a closure test for our method. Note that the small differences in the fragmentation function for different weights, compare to fig. 5, combine into considerable differences for complete fragmentation histories, as shown by the relatively larger differences between the $-2\ln w_{\text{exact}}$ distributions given in the right panel of fig. 19. These differences, however, do not translate to large differences of event-level weights, as shown by the $-2\ln w_{\text{class}}$ distributions in the left panel of fig. 19.

---

[14]The definitions of the high-level observables are given in Appendix B of ref. [3].

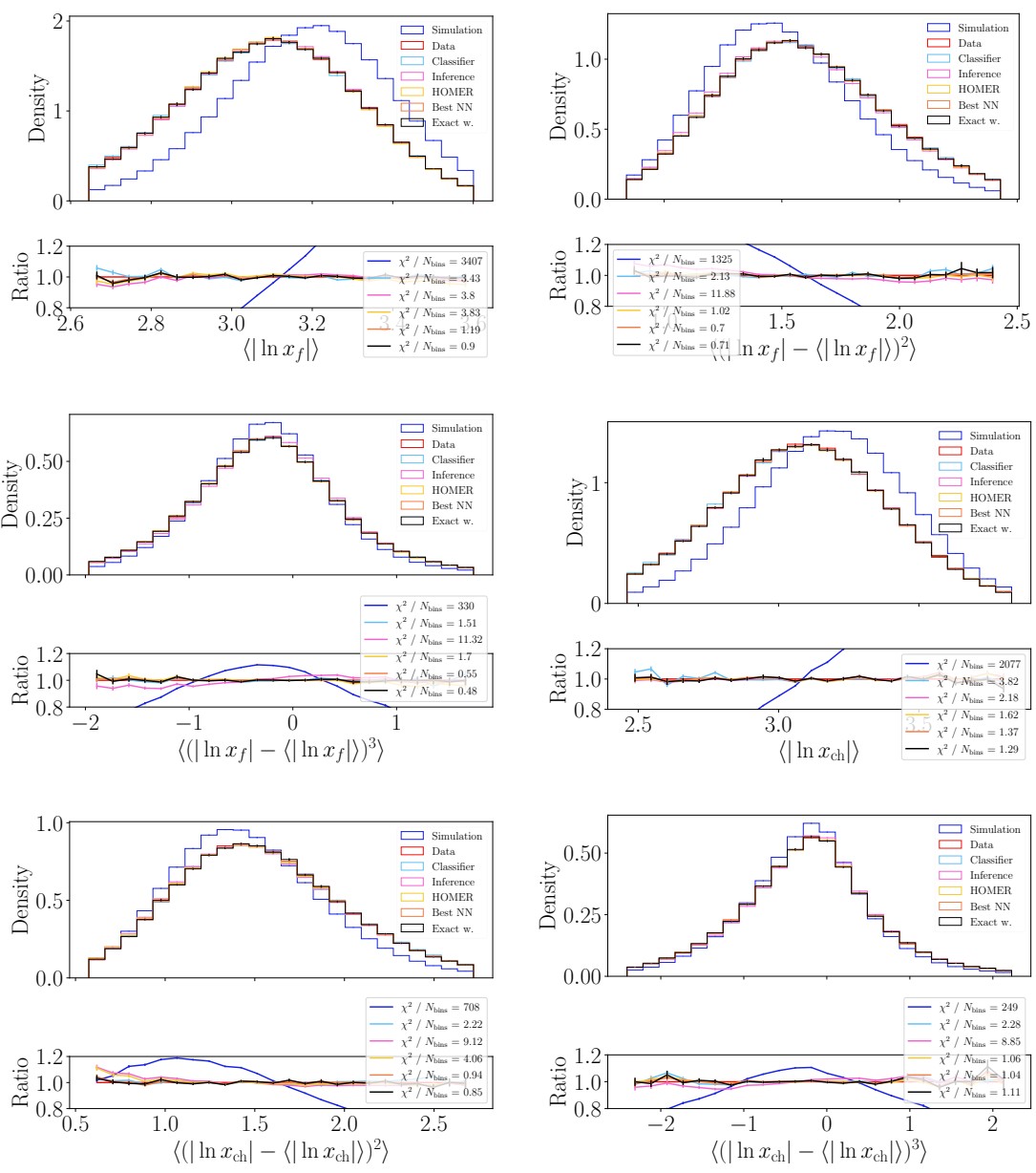

Figure 17: Reweighted distributions of high-level observables for the fixed $qg\bar{q}$ scenario. All weights originate from the model trained with unbinned high-level observables.

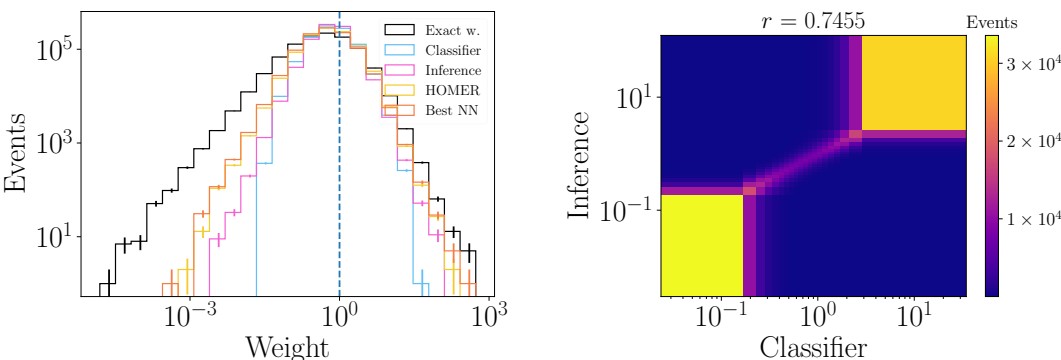

Figure 18: Comparison of weights obtained using the unbinned high-level observables for the fixed $qg\bar{q}$ scenario: (left) all event weights and (right) comparison between the $w_{\text{class}}$ weights obtained in Step 1 and the inferred weight estimators $w_{\text{infer}}$ from Step 2.

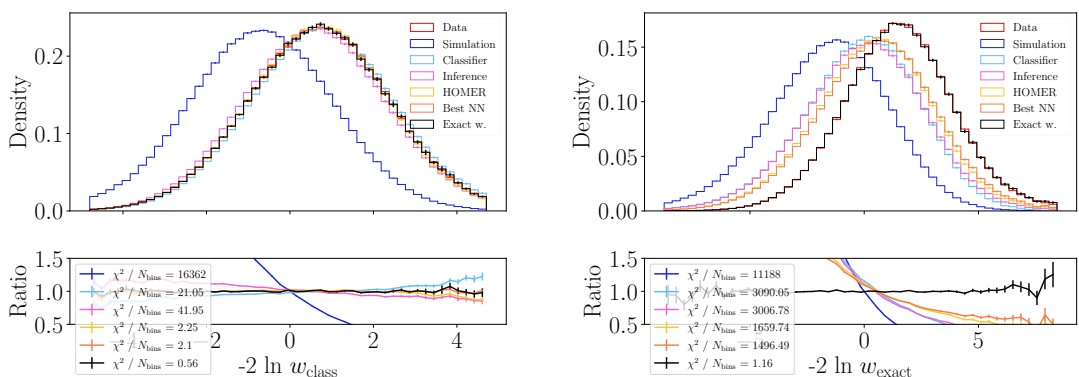

Figure 19: Reweighted distributions from the fixed $qg\bar{q}$ scenario for (left) the output of the Step 1 classifier and (right) for the optimal observable obtained from the exact weights computed from PYTHIA. All model weights are from the model trained with unbinned high-level observables.

## B.2   Variable $qg\bar{q}$ scenario

In this subsection we collect additional figures for the variable $qg\bar{q}$ scenario. These results supplement those already discussed in section 4.4.2.

Figure 20 shows the distributions for the high-level observables $1 - T$, $C$, $D$, $B_W$, and $B_T$. Similarly, fig. 21 shows the predictions for the moments of $\ln x_f$ and $\ln x_{\text{ch}}$ distributions, to be compared with the results of fig. 17 for the fixed $qg\bar{q}$ scenario. Much like the results for the fixed $qg\bar{q}$ scenario, these results show a percent-level agreement between the HOMER predictions and data, including the intermediate HOMER results. Similarly, the goodness-of-fit statistic, $\chi^2/N_{\text{bins}}$, is comparable between the *Classifier* and *HOMER* distributions, and only somewhat larger than those obtained with the *Best NN* and *Exact weights*.

Similar conclusions to those of appendix B.1 can be drawn here for the distributions of weights and the correlations between $w_{\text{class}}$ and $w_{\text{infer}}$, shown in fig. 22. The difference between the $w_{\text{class}}$ and $w_{\text{class}}$ distributions is still relatively minor, but does show an increased information gap compared to the fixed $qg\bar{q}$ scenario of fig. 18. This increased information gap also translates to larger differences between the distributions for the optimal statistic, shown

in fig. 23, to be compared with the distributions of fig. 19 for the fixed $qg\bar{q}$ scenario. Note that the event-level optimal statistic, $-2\ln w_{\mathrm{class}}$, shown in the left panel of fig. 23, still follows the data quite closely, particularly for the final HOMER results obtained using $w_{\mathrm{HOMER}}$. However, the *Classifier*, and especially *Inference* distributions deviate more appreciably from data then they did in the simpler fixed $qg\bar{q}$ scenario.

Figure 20: Distributions of high-level observables obtained at different stages of the HOMER method for the variable $qg\bar{q}$ scenario. The corresponding multiplicity distributions are given in fig. 9.



Figure 21: Reweighted distributions of high-level observables for the variable $qg\bar{q}$ scenario. All weights originate from the model trained with unbinned high-level observables.

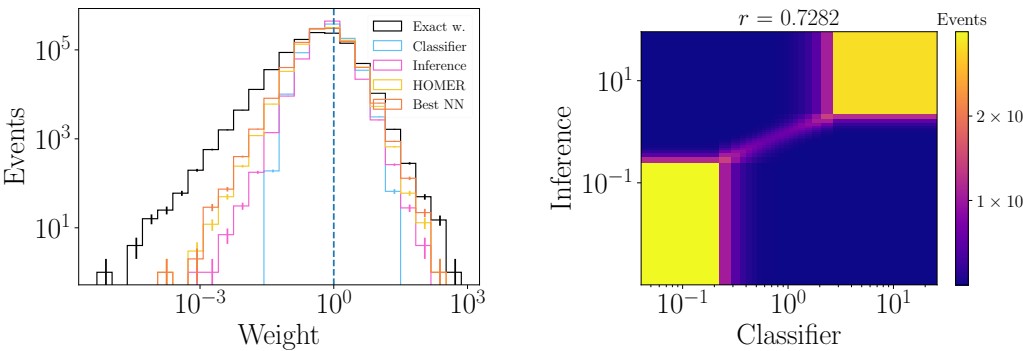

Figure 22: Comparison of weights obtained using the unbinned high-level observables for the variable $qg\bar{q}$ scenario: (left) all event weights and (right) comparison between the $w_{\text{class}}$ weights obtained in Step 1 and the inferred weight estimators $w_{\text{infer}}$ from Step 2.

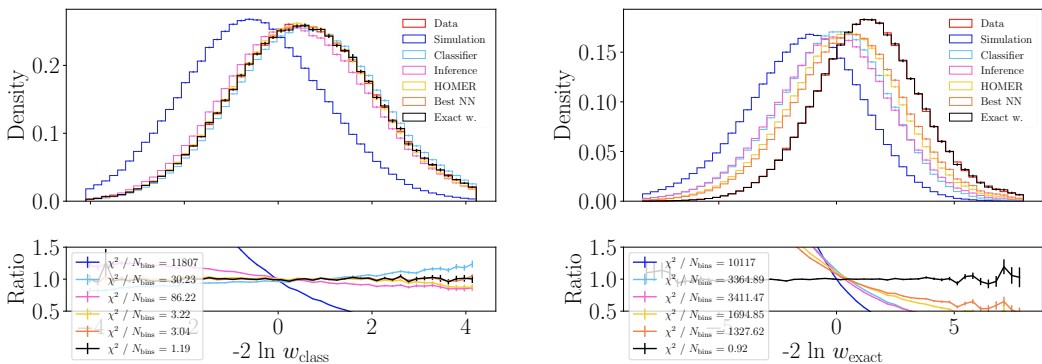

Figure 23: Reweighted distributions from the variable $qg\bar{q}$ scenario for (left) the output of the Step 1 classifier and (right) for the optimal observable obtained from the exact weights computed from PYTHIA. All model weights are from the model trained with unbinned high-level observables.

## B.3  Variable $qg^{(n)}\bar{q}$ scenario

Here we collect additional results for the variable $qg^{(n)}\bar{q}$ scenario. These results supplement the ones shown in section 4.4.3 in the main text, and should also be compared to similar results for the fixed $qg\bar{q}$ scenario of appendix B.1 and the variable $qg\bar{q}$ scenario of appendix B.2.

The distributions for high-level observables $1 - T$, $C$, $D$, $B_W$, and $B_T$ are shown in fig. 24, to be compared with figs. 16 and 20 for the $qg\bar{q}$ scenarios. The predictions for the moments of the $\ln x_f$ and $\ln x_{\text{ch}}$ distributions are shown in fig. 25, to be compared with the results of figs. 17 and 21.

Here, there is a degradation in the fidelity of HOMER predictions for all observables, which can be traced to a lower quality of Step 1 results, *i.e.*, to the performance of the Step 1 classifier, and the fidelity of the resulting $w_{\text{class}}$ weights. The lower effective statistics of the datasets that are used in the training, which are the result of higher variance in the probabilities for fragmentation chains that lead to the same events, also result in further degradation, see table 1 and table 2. Consequently, the performance of Step 2 in the HOMER method also is significantly degraded, compared to the case of strings with a single gluon. This is, for instance, signaled by the significantly lower Pearson correlation coefficient of fig. 26, compared to the two cases with only one gluon in figs. 18 and 22.



Figure 24: Distributions of high-level observables obtained at different stages of the HOMER method for the variable $qg^{(n)}\bar{q}$ scenario. The corresponding multiplicity distributions are given in fig. 11.

The overall degradation in performance of the HOMER predictions is less important for the observables that are most sensitive to hadronization, such as the multiplicities of fig. 11. For instance, while the goodness-of-fit value $\chi^2/N_{\rm bins} \sim 10$ for the HOMER predictions for $\ln x_f$ is much larger than for the exact weight distribution, $\chi^2/N_{\rm bins} \sim 1$, it is still more than an order of magnitude smaller than the difference between data and simulation. That is, the HOMER prediction for $\ln x_f$ follows the correct distribution at the level of a few percent in the bulk of the distribution; for $\ln x_{\rm ch}$ the fidelity of the HOMER prediction is even better. This trend is much less pronounced for the observables that are less sensitive to hadronization effects, or, more precisely, in the change of the $a$ parameter in the Lund string model, such as the high-level shape observables, see fig. 24. Since for these the difference between *Simulation* and *Data* distributions is relatively small, the HOMER predictions are sometimes as far from *Data* distributions as the *Simulation* distributions.



Figure 25: Reweighted distributions of high-level observables for the variable $qg^{(n)}\bar{q}$ scenario. All weights originate from the model trained with unbinned high-level observables.

The left panel of fig. 26 shows the distribution of weights, to be compared with the results of figs. 18 and 22. One can see that there is a larger difference between the $w_{\text{class}}$ and $w_{\text{exact}}$ distributions than found for the fixed and variables $qg\bar{q}$ scenarios, signaling an increased gap between the information available at the fragmentation level with $w_{\text{exact}}$ and event level with $w_{\text{class}}$.

The distributions for the optimal statistics are shown in fig. 27, to be compared with the distributions of figs. 19 and 23, and also show a decreased performance of the HOMER predictions. However, the HOMER prediction for $-2\ln w_{\text{class}}$ is not very far from the *Best NN* results, *i.e.*, for NNs trained directly on the fragmentation function weights. This signals that, although the underlying function is imperfect as exemplified in the predicted $-2\ln w_{\text{exact}}$ distribution, the high-level observables are still well modeled.

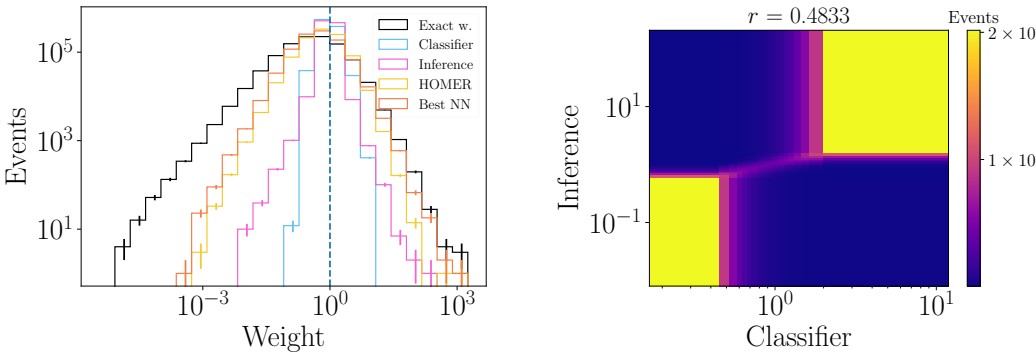

Figure 26: Comparison of weights obtained using the unbinned high-level observables for the variable $qg^{(n)}\bar{q}$ scenario: (left) all event weights and (right) comparison between the $w_{\text{class}}$ weights obtained in Step 1 and the inferred weight estimators $w_{\text{infer}}$ from Step 2.

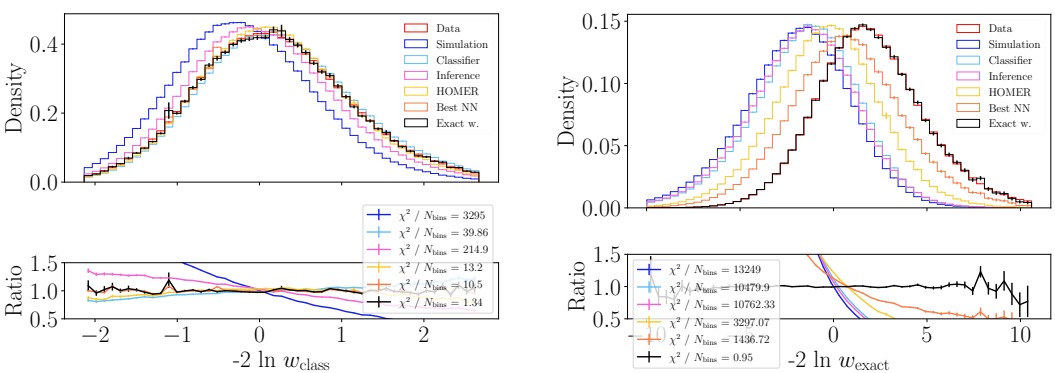

Figure 27: Reweighted distributions from the variable $qg^{(n)}\bar{q}$ scenario for (left) the output of the Step 1 classifier and (right) for the optimal observable obtained from the exact weights computed from PYTHIA. All model weights are from the model trained with unbinned high-level observables.

## C  An example with both $a$ and $b$ changed

In this appendix we show the results of HOMER when fitting to a dataset with variable $qg^{(n)}\bar{q}$ generated by changing both $a$ and $b$ from $(0.68, 0.98)$ to $(0.55, 0.78)$. The specific parameter values are selected to ensure sufficient coverage and to assess whether HOMER remains robust under more arbitrary modifications, while still serving as a valid closure test.

In fig. 28 we present the values of $\chi^2(\mathcal{O}, \sigma_s)/N_{\text{bins}}$, as defined in eq. (20). These results are consistent with those in fig. 4, where the optimal choice of $\sigma_s$ reflects a trade-off between Step 2 performance and generalization capability. This behavior is further illustrated in fig. 33, where the left panel shows the distribution of weights and the right panel compares the performance between Step 1 and Step 2 for the best-performing $\sigma_s^*$.

Figure 29 shows the Lund string fragmentation function that has been extracted using the HOMER method. The right panel in fig. 29 shows $f(z)$ for a particular transverse mass bin, $m_T^2 \in [0.066, 0.095)$ GeV$^2$, while the left panel in fig. 29 gives the value of $f(z)$ averaged over all $m_T^2$ bins. In both cases, the results were also averaged over all the remaining variables. We see that the extracted fragmentation function, although not perfect, represents an improvement over the reference distribution.

The distributions of high-level observables obtained using the optimal smearing, $\sigma_s^*$, are shown in figs. 30 to 32. These results show that HOMER achieves performance comparable to that of Step 1.

Finally, the distributions for the optimal statistics are shown in fig. 34. We observe that the performance is similar to the one obtained in the main text, i.e., for single parameter variation example, showcasing the flexibility of HOMER.

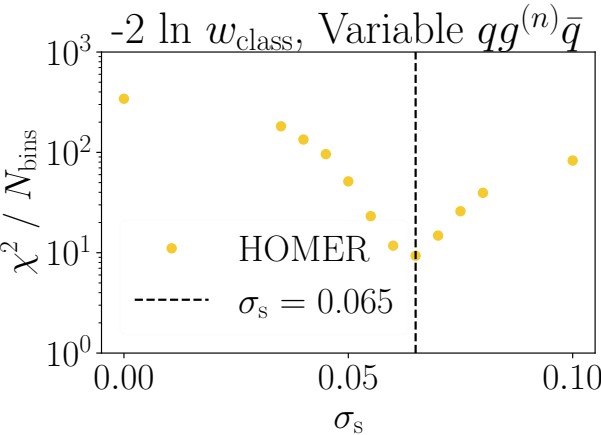

Figure 28: Goodness-of-fit defined by eq. (20), shown as a function of $\sigma_s$ for the alternative "Data" with $N_{\text{bins}} = 50$.

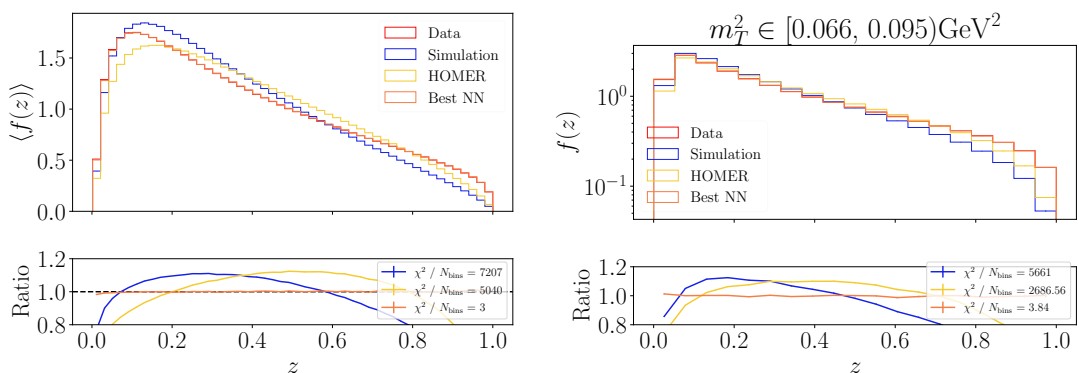

Figure 29: (left) Reweighted distributions for the fragmentation function averaged over all string break variables except $z$ and (right) fixing the transverse mass bin. All weights are from a model trained with the unbinned high-level observables for the variable $qg^{(n)}\bar{q}$ and alternative "Data" scenario.

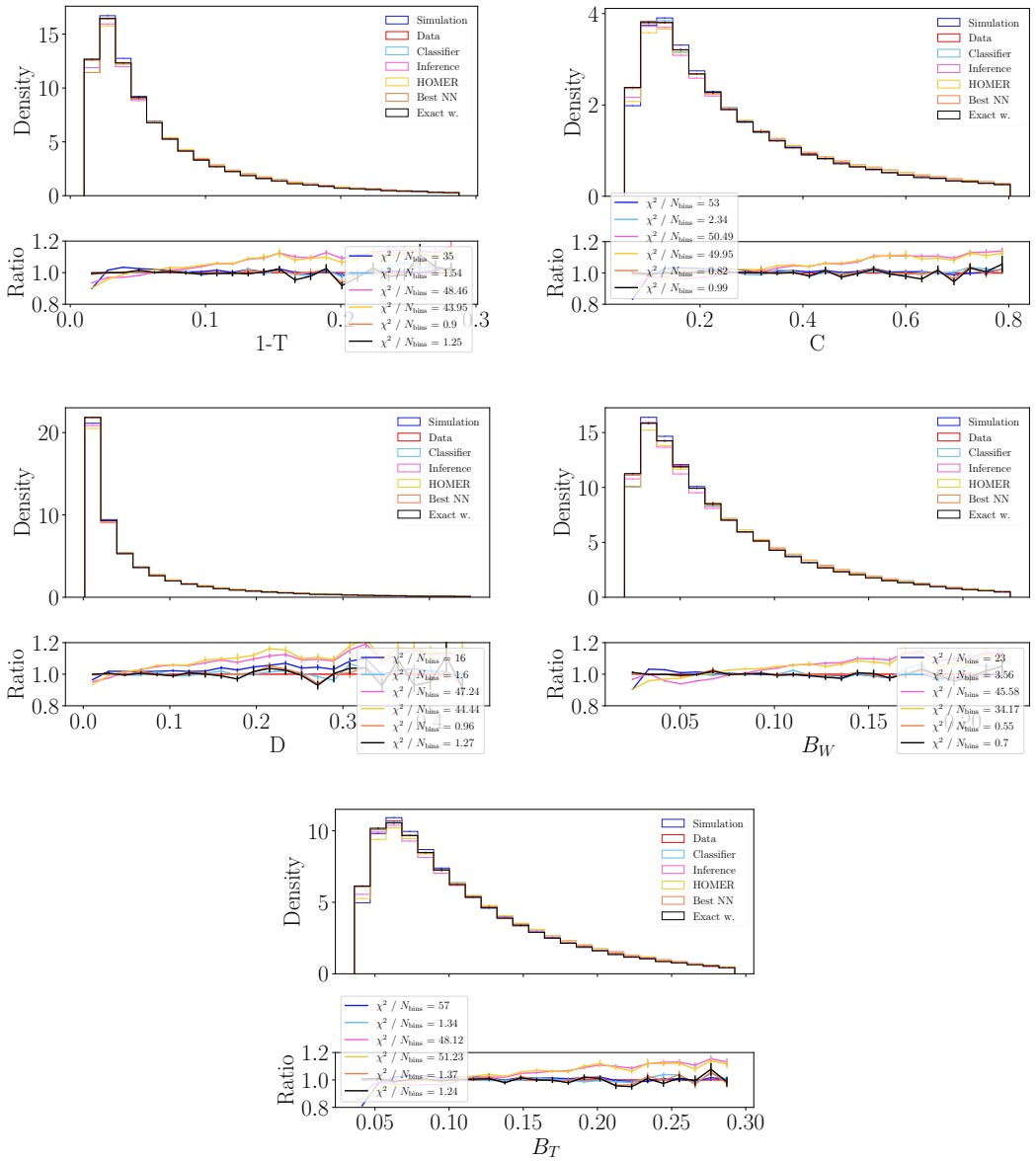

Figure 30: Distributions of high-level observables obtained at different stages of the HOMER method for the variable $qg^{(n)}\bar{q}$ and alternative "Data" scenario. The corresponding multiplicity distributions are given in fig. 31.

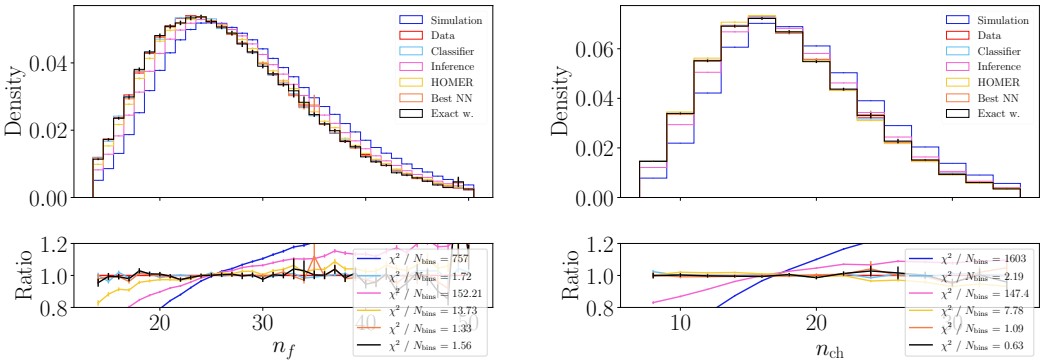

Figure 31: (left) Multiplicity and (right) charged multiplicity distributions for the variable $qg^{(n)}\bar{q}$ and alternative "Data" scenario, where the training was performed on unbinned high-level observables.

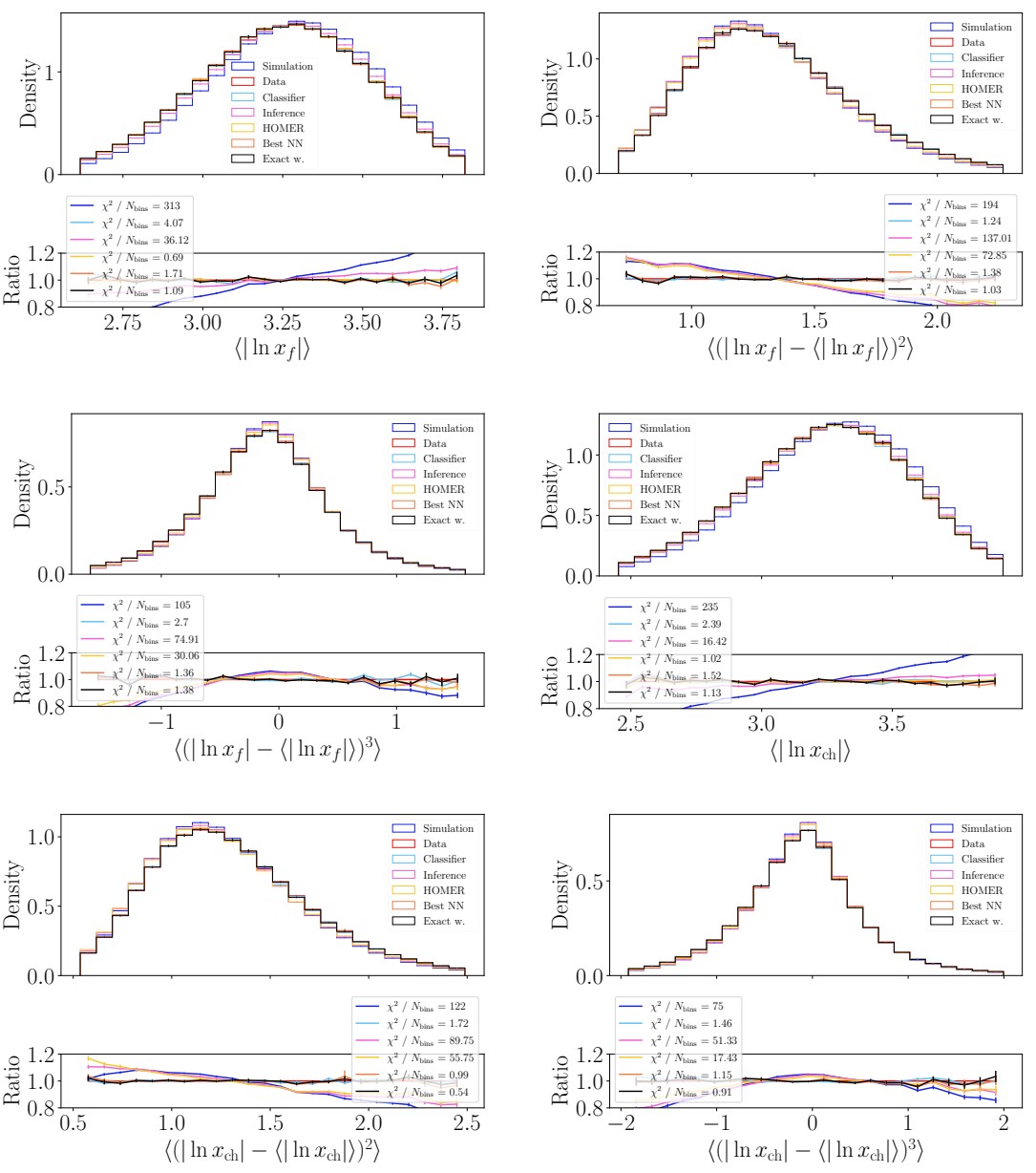

Figure 32: Reweighted distributions of high-level observables for the variable $qg^{(n)}\bar{q}$ and alternative "Data" scenario. All weights originate from the model trained with unbinned high-level observables.

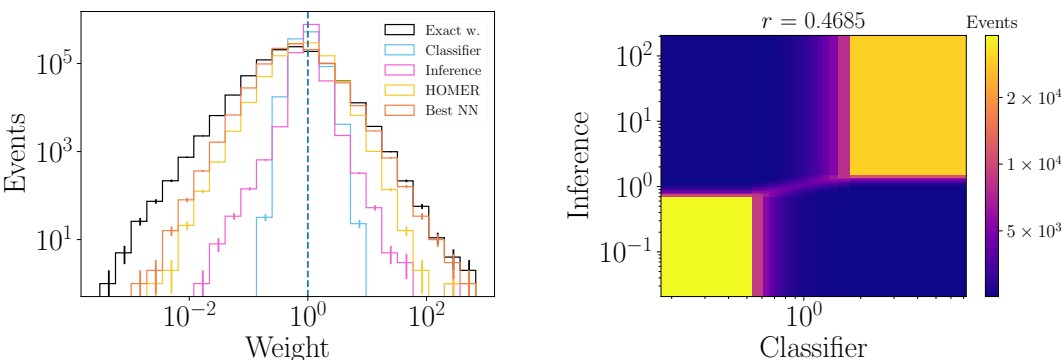

Figure 33: Comparison of weights obtained using the unbinned high-level observables for the variable $qg^{(n)}\bar{q}$ and alternative "Data" scenario: (left) all event weights and (right) comparison between the $w_{\text{class}}$ weights obtained in Step 1 and the inferred weight estimators $w_{\text{infer}}$ from Step 2.

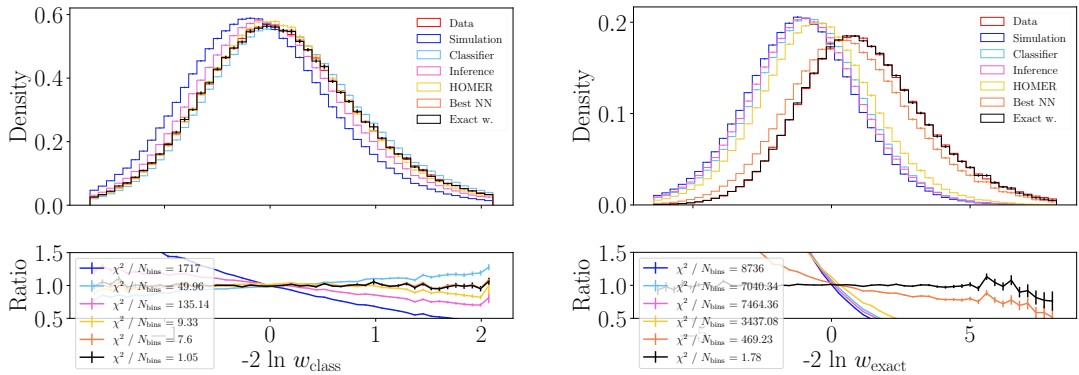

Figure 34: Reweighted distributions from the variable $qg^{(n)}\bar{q}$ and alternative "Data" scenario for (left) the output of the Step 1 classifier and (right) for the optimal observable obtained from the exact weights computed from PYTHIA. All model weights are from the model trained with unbinned high-level observables.

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
