# Peer review of "Characterizing the hadronization of parton showers using the HOMER method"

_SciPost Physics, doi:SciPost Phys. 19, 125 (2025)_

## Round 1 · Referee Report · Anonymous (Referee 2) · 2025-8-3

Report
The results shown in this impressive paper are a key step forward from the original HOMER paper, towards being able to apply this method on actual data. Let me emphasize, first, that I consider HOMER an excellent attempt to improve hadronization models for the LHC, without losing its physics interpretability to a fully flexible but not theory-motivated parametrization. So the paper should definitely be published, and there are only a few minor questions and comments I have:
1- I am not fully convinced that the current structure of Sec.2 is the best possible. Personally, would first have reviewed the quark-only HOMER structure, including Fig.2, and then introduced the additional gluon issues, as maybe part of Sec.3. But of course I am only a reader, not the author of the paper.
2- In essence, Sec.3.1 is too brief to underststand it, and too long to just skip it. Maybe a graphic representation of some kind of help? I like Fig.3, and it would be even more useful if it were complemented with a graphic representation of step 2 in the original paper? Even though it might appear trivial?
3- The role of exact weights becomes only really obvious in the later part of the paper, maybe this aspect can be introduced in Sec.3 in a more targeted manner?
4- I am sorry, but Sec.3.2.3 seems technically very expensive, as is said in the text. It might help to discuss simpler alternatives and why they do not work or why the authors still choose this way. This is related to a question concerning 4.1, namely why the authors use such a simple GBC for Step 1 and such an advanced MPGNN for Step 2. Please explain that range of choices.
5- As described in 4.1, why is only one parameter changed in the `data' setup? Would it not be more natural to make and test more adjustments?
6- In Sec.4.3 the authors give numbers for \sigma_s^*, what is the range or error on this choice? Similarly, for instance in Tab.3 I would be happier if the authors could assign an uncertainty to the numbers. How stable are these ratios?
7- The figures of the paper could be improved significantly. For instance I would prefer to see all four scenarios of Fig.4 in one plot with different symbols. In Fig.5 some of the text is very small.
8 - Concerning Fig.7, are the assumptions of the SHAP analysis justified, especially the de-correlation assumption which is normally behind the SHAP implementation?
9- I personally get annoyed by footnotes, as they interrupt the text, and the footnotes in this paper could just be included in the text.
10- Layout etc: E_cm below Eq.(2) is missing an opening parenthesis, therefor should be therefore; some appearances for instance of finalTwo does not commute with good line breaks.
Some of these are just questions or comments from my side, so I do not require them to be followed for the paper. But they might help making a very nice and totally publishable paper even nicer.
Recommendation
Ask for minor revision
Strengths
1- clearly written 2- high-quality supporting plots (although some of them could be a bit larger!) 3- useful appendices
Weaknesses
1- none, really
Report
Recommendation
Publish (easily meets expectations and criteria for this Journal; among top 50%)
We thank the referee for its very positive review.

Author: Manuel Szewc on 2025-09-24 [id 5862]
(in reply to Report 2 on 2025-08-03)We thank the referee for the careful reading of our manuscript and for the constructive feedback. Below we address each point in turn.
The results shown in this impressive paper are a key step forward from the original HOMER paper, towards being able to apply this method on actual data. Let me emphasize, first, that I consider HOMER an excellent attempt to improve hadronization models for the LHC, without losing its physics interpretability to a fully flexible but not theory-motivated parametrization. So the paper should definitely be published, and there are only a few minor questions and comments I have:
We appreciate that the suggested reorganization could be more pedagogically accessible. We believe, however, that the current organization allows for the greatest utility as a reference while minimizing redundancy with our previous work. We have thus opted not to reorganize Sec.2.
We have added a reference to Figure 2 of the original paper. As with the previous response, we think this solution strikes a good balance between pedagogical and reference utility while avoiding redundancy.
Thank you for the suggestion. We have added a sentence to the second paragraph of section 3.1 to clarify. We have also clarified the relationship between $w_\mathrm{exact}$ (introduced in section 4.2) and the "exact weights" (introduced in section 4.4) in section 4.4.
4- I am sorry, but Sec.3.2.3 seems technically very expensive, as is said in the text. It might help to discuss simpler alternatives and why they do not work or why the authors still choose this way. This is related to a question concerning 4.1, namely why the authors use such a simple GBC for Step 1 and such an advanced MPGNN for Step 2. Please explain that range of choices.
The referee has nothing to apologize for, it is true that the smearing is expensive even for a fixed smearing parameter. However, we found it manageable, and necessary to perform the averaging over multiple initial states and over compatible accepted chains, as explained in the text. We are not aware of simpler alternatives, but are exploring other alternatives which may be more complex in implementation but perhaps scale better. We have added a clarifying sentence to this regard at the end of Section 3.2.2.
Regarding the choice of a MPGNN for Step 2, this is because we want a fast, established implementation that can deal with variable size point clouds. Conversely, the choice of GBC is made because the high-level event-by-event observables allow for it, and it is simple, fast and powerful. We found these to be the simplest choices for each data representation. We have clarified this point further in Section 4.1.
The single parameter change was chosen so that the problem is simple to perform a closure test on, and also to validate and compare against previous work (Refs. HOMER and reweighting). However, the reviewer is correct in stating that nothing forbids more involved modifications, provided the reference distribution is close enough that we can reweight from it. To demonstrate this we have now included in a new appendix the results for simulaneous modifications of both a and b parameters. The fidelity of the results is similar to the case where only a parameter was varied.
We agree with the referee that assigning errors would be a worthwhile endeavor and it will be the subject of future work. However, this is also not an easy task, and requires significant extension of present work: even the simplest approach to this problem, such as studying the variation over multiple initializations of the \stepTwo neural networks, would require a large number of additional runs, making the task burdensome without further developments, and unfortunately beyond the scope of present work. Empirically, we rely on large enough samples, such that we are rather confident in the stability of the results, though at the moment we cannot prove this definitively, nor assign a reliable error estimate without more involved study. To make this clear to the reader, we have added a sentence to this effect in the conclusions.
We agree and have made the following improvements: we have collected all Fig. 4 plots into a single figure and have updated the font size of the fragmentation function plots accordingly.
This is a very relevant question. The SHAP values are sufficient for our purposes, because we are using a Boosted Decision Tree as a classifier. In this case, the SHAP value computation does not assume the features to be de-correlated (see Ref. https://www.nature.com/articles/s42256-019-0138-9). However, this also does not imply that the SHAP values contain all of the relevant information, since they still take one observable at a time. For instance, evaluation of feature interactions is needed to better establish the effect of multiple feature variations. We have added a short discussion as a footnote in Section 4.4.1.
We understand that there are different views on the use of footnotes. While we try to minimize the use of footnotes as a general rule, the ones that we have kept in the manuscript do, in our opinion, help with the flow of the paper. We have thus opted not to make any changes to the footnotes, hoping that the referee will understand different approaches to the writing styles.
We thank the referee for catching these errors, which we have corrected in the updated draft.

---

## Round 3 · Author Response

List of changes
- We have fixed the typos pointed out by the reports.
- We have expanded on the definition of exact weights in Section 3.1 and clarified their relationship to the weights detailed in Section 4.4.
- We have added a clarifying sentence regarding the numerical costs of the introduction of a smearing kernel at the end of Section 3.2.2.
- We have clarifited our choice of GBC and MPGNN in Section 4.1.
- We have clarified the assumptions behind the use of Shapley values in Section 4.4.1.
- We have modified multiple figures to increase readability.
- We have added a sentence in the conclusions regarding the need for uncertainty quantification in the determination of the smearing kernel width.
- We have included an additional appendix where we explore a HOMER fit to data generated with two simultaneous parameter variations.

---

## Round 3 · List of Changes

- We have fixed the typos pointed out by the reports.
- We have expanded on the definition of exact weights in Section 3.1 and clarified their relationship to the weights detailed in Section 4.4.
- We have added a clarifying sentence regarding the numerical costs of the introduction of a smearing kernel at the end of Section 3.2.2.
- We have clarifited our choice of GBC and MPGNN in Section 4.1.
- We have clarified the assumptions behind the use of Shapley values in Section 4.4.1.
- We have modified multiple figures to increase readability.
- We have added a sentence in the conclusions regarding the need for uncertainty quantification in the determination of the smearing kernel width.
- We have included an additional appendix where we explore a HOMER fit to data generated with two simultaneous parameter variations.

---

## Editorial Decision

published